# Simulation of Long-Term Soil Hydrological Conditions at Three Agricultural Experimental Field Plots Compared with Measurements

**Martin Wegehenkel** [1,*]**, Karin Luzi** [2]**, Dieter Sowa** [3]**, Dietmar Barkusky** [4] **and Wilfried Mirschel** [2]

[1] Leibniz Centre for Agricultural Landscape Research ZALF, Research Area 2 "Land Use and Governance", Working group "Lowland Hydrology und Water Management", 15374 Müncheberg, Germany

[2] Centre for Agricultural Landscape Research ZALF, Research Platform "Models and Simulation", Working Group "Integrated Landscape Modeling", 15374 Müncheberg, Germany; kluzi@zalf.de (K.L.); wmirschel@zalf.de (W.M.)

[3] Centre for Agricultural Landscape Research ZALF, Research Platform "Models and Simulation", Working Group "Ecosystem Modelling", 15374 Müncheberg, Germany; Dieter.Sowa@zalf.de

[4] Centre for Agricultural Landscape Research ZALF, Experimental Infrastructure Platform, Working Group "Experimental Station Müncheberg", 15374 Müncheberg, Germany; dbarkusky@zalf.de

* Correspondence: mwegehenkel@zalf.de

**Abstract:** Soil hydrological conditions influence crop growth and groundwater recharge and, thus, precise knowledge of such conditions at field scale is important for the investigation of agricultural systems. In our study, we analyzed soil hydrological conditions at three agricultural experimental field plots with sandy soils and different crop rotations using a 22-year period from 1993 to 2014 with daily volumetric soil water contents measured by the Time Domain Reflectometry with Intelligent MicroElements (TRIME)-method and pressure heads determined by automatic recording tensiometers. These measured data were compared with soil water contents and pressure heads simulated by a process-based agroecosystem model. Within this 22-year period, time spans with a better model performance and periods with a lower goodness of fit between simulations and observations were observed. The lower goodness of fit in the summer periods was attributed to inadequate calculations of root water uptake. Measurement errors of the TRIME-probes and differences between soil water contents measured by TRIME and pressure heads observed by tensiometers due to different measurement volumes, precision and measuring principles were identified as further reasons for mismatches between simulated and measured model outputs.

**Keywords:** soil water balance modelling; model validation; TRIME; tensiometer; measurements

## 1. Introduction

In the field of agricultural science, soil moisture influences the germination of seeds, plant growth and plant nutrition, microbial decomposition of soil organic matter, nutrient transformations in the root zone as well as heat and water transfer at the crop-atmosphere interface [1]. In addition, the efficiency of irrigation management practices depends on an accurate in situ estimation of temporal soil moisture dynamics in the root zone. Therefore, measurement and modelling of soil moisture is important in hydrology and agriculture [2,3].

Thus, computer codes for the simulation of soil water fluxes are important parts of process-based agroecosystem models, which simulate, e.g., soil hydrology and crop growth. Such models are used to predict, e.g., the impact of different farm management practices, alternative cropping systems, climate

change and drought on, e.g., runoff, erosion, evapotranspiration, seepage, soil water- and nutrient balance and crop growth [4–7]. However, simplified assumptions about the processes and parameters of soil water retention and soil water fluxes might make model predictions uncertain. Thus, for an estimation of the accuracy of the predictions of such models, a thorough validation by comparing in situ measured and calculated state variables such as soil water contents and pressure heads is important. Such a validation requires coherent, continuous long-term time series with a high temporal resolution of these measured state variables, especially if these models are used for the estimation of the impact of climate change [4–9]. Such estimates are based on input data obtained from climate change scenarios with time spans sometimes longer than 50 years [4–9].

For an automatic non-destructive measurement of soil water contents and pressure heads at field scale, the application of Time-Domain-Reflectometry (TDR) and Frequency Domain Reflectometry (FDR) as well as automatic recording tensiometers was established in the last decades [10–12]. Such measurement techniques can yield the required long-term observation periods with a high time resolution and, thus, can determine the wet and dry extremes of soil hydrological conditions. Accurate knowledge is required of both the wet extremes because of their important contribution to soil water flow and soil water storage, and the dry extremes in order to validate models of root water uptake. Several recent studies postulate a lack of long-term, multiple-year, continuous and quality checked measurements of soil hydrological conditions with high temporal resolution for the previous mentioned thorough validation of process-based agroecosystem models and corresponding model validation studies to check the quality and consistency of the model predictions for soil water balance over long-time periods [7–9,13–15].

Thus, the overall objective of our study was an analysis and modelling of long-term soil moisture dynamics at three agricultural experimental field plots. In this regard, the following specific objectives were addressed:

- Simulation of long-term soil moisture dynamics and thorough validation of a process-based agroecosystem model using long-term consistent and continuous time series of daily soil water contents measured by TDR and pressure heads observed by tensiometers and an analysis of model performance.
- Identification of reasons for mismatches between simulated and measured soil water contents and pressure heads with respect to seasonal effects, hydrometeorological conditions, soil hydraulic parameters, applied measurement techniques, cultivated crop types and model assumptions.

Our 22 year dataset covered several vegetation periods with different crops and a wide range of wet, dry, cool and warm years and, thus, offer new opportunities for, e.g., model parametrization, model validation, analysis of soil water extraction by vegetation cover and model comparison as compared to other recent studies using similar measurement set ups with a focus on model validation with distinct shorter time periods and with a lower variation in vegetation or crop cover [13,16–24].

## 2. Materials and Methods

### 2.1. Simulation Model

In our study, we used the modelling system THESEUS [25]. In THESEUS, crop growth is simulated using algorithms obtained from the Wofost7.1 model [26]. These algorithms calculate phenological development, $CO_2$-assimilation, root water uptake (RWU), growth and maintenance respiration, distribution of assimilates on stem, leaf, fruit and root as well as dry matter formation. Daily potential evapotranspiration $ET_P$ (mm day$^{-1}$) is calculated using a modified Penman-approach [27,28] as

$$ET_P = \frac{\Delta \cdot R_{na} + \gamma \cdot (0.26 \cdot (e_s - e_a) \cdot (f + c \cdot u(2))}{\Delta + \gamma}, \tag{1}$$

where $\Delta$ is the slope of the saturation vapor pressure curve in kPa $°C^{-1}$; $R_{na}$ is the net radiation defined as evapotranspiration equivalent in mm $day^{-1}$; $\gamma$ is the psychrometer constant = 0.65 in kPa $°C^{-1}$; $e_s$ is the saturated vapor pressure and $e_a$ is the actual vapor pressure, both in kPa; f is an empirical constant = 1.0 and c is an empirical coefficient calculated from the difference between daily maximum and minimum air temperature; u(2) is the mean windspeed at 2 m height in m $s^{-1}$.

Soil evaporation is restricted to the soil surface layer and is limited by soil water content at air dryness. Leaf area index (LAI) is the major determinant for light absorption and photosynthesis of the crop and for partitioning of $ET_P$ in potential soil evaporation $E_{pot}$ and potential RWU $T_{pot}$ according to [29].

$$E_{pot} = \exp(-\text{ext\_coeff} \cdot \text{LAI}) \cdot ET_P \text{ and } T_{pot} = [1 - \exp(-\text{ext\_coeff} \cdot \text{LAI})] \cdot ET_P, \tag{2}$$

where ext_coeff is a crop specific extinction coefficient of radiation [29]. Thus, the crop growth model simulates growth of LAI with a high level of detail. In contrast, rooting depth is simulated in a more simplified manner. After initialization, the crop grows with a fixed daily increase in rooting depth until either a crop-specific maximum depth or a soil-defined maximum depth is reached or if there is no partitioning of biomass to roots anymore. Crop growth is limited by soil water availability for RWU, temperature stress and oxygen shortage due to stagnant water in the root zone [26]. The distribution of soil water extraction by RWU between soil surface and simulated actual rooting depth L is calculated according to [30].

$$g(i) = \frac{(c+1) \cdot \left(\frac{L \cdot c - x}{L \cdot c + x - 1}\right) \cdot \frac{1}{L}}{(c+1) \cdot \ln\left(\frac{c+1}{c}\right) - 1}, \tag{3}$$

where g(i) is the fraction of RWU extracted from soil layer i, x is the difference between soil surface and the soil layer i in dm and c is a parameter for the position of water extraction in the soil profile. The higher the value of c, the higher is RWU in the upper soil layers. In our study we used a value of c = 10 [30]. The impact of soil water availability on RWU at each soil layer i is calculated as:

$$RFWS_i = \frac{\theta_{a,i} - \theta_{wp,i}}{\theta_{cri,i} - \theta_{wp,i}} \quad \theta_{cri,i} = (1-p) \cdot \left(\theta_{fc,i} - \theta_{wp,i}\right) + \theta_{wp,i}. \tag{4}$$

Here, $RFWS_i$ is the dimensionless reduction factor for RWU from 0 to 1, $\theta_{a,i}$ is actual soil water content in soil layer i, $\theta_{wp,i}$ is soil water content at wilting point, $\theta_{fc,i}$ is soil water content at field capacity, $\theta_{cri,i}$ is critical soil water content defined as the threshold below which RWU is reduced. All these values are in $cm^3$ $cm^{-3}$. The soil water depletion fraction p is a function of $ET_P$ and depends on the drought sensitivity of the crop. Values of the ratio actual versus potential RWU < 1 reduce gross assimilation and, thus, crop growth.

The simulation of soil water fluxes is based on a numerical solution of the flux density and continuity equations [25,31]. The flux density q in cm $day^{-1}$ is calculated as

$$q = K(h) \cdot \frac{\partial H}{\partial z} \text{ with } H = h + z, \tag{5}$$

where H is the hydraulic head composed of soil water pressure head h and gravitational head z, both in cm; K(h) is the soil hydraulic conductivity in cm $day^{-1}$ and z is the depth in cm. Changes in soil water contents per time step $\partial t$ are obtained from the continuity equation.

$$\frac{\partial \theta}{\partial t} = \frac{\partial q}{\partial z} + s, \tag{6}$$

where s is the sink term in cm $d^{-1}$ and $\theta$ is the soil water content in $cm^3$ $cm^{-3}$. The soil water retention $\theta(h)$ and hydraulic conductivity functions K(h) were described by the equations according to [32,33].

$$K(h) = \frac{K_{sat}(1-(\alpha|h|)^{n-1}\left[1+(\alpha|h|)^n\right]^{-m})^2}{(1+(\alpha|h|)^n)^{\frac{m}{2}}} \text{ and } \theta(h) = \theta_r + \frac{\theta_s - \theta_r}{(1+(\alpha|h|)^n)^m},\quad (7)$$

where $K_{sat}$ is the saturated hydraulic conductivity in cm day$^{-1}$; $\alpha$ in cm$^{-1}$ and n without dimensions are soil-specific parameters with the restriction m = 1− 1/n; $\theta_s$ is the saturated and $\theta_r$ is the residual soil water content. These parameters are referred to herein as vGM-parameters.

## 2.2. Description of the Test Site

The Experimental Station of the Leibniz Center for Agricultural Landscape Research (ZALF), Müncheberg, Germany is located around 50 km east of Berlin (52°52′ N, 14°07′ E, 62 m.a.sl.)

At three field plots (Figure A1), measurement systems were installed for long-term monitoring for soil hydrological conditions in terms of soil water contents and pressure heads under rain-fed conditions. The dataset consists of daily precipitation, global radiation, maximum and minimum air temperature, wind speed, saturation deficit of air and soil temperatures measured by an automatically recording weather station located in the Northwestern part of the field plots (Figure A1), soil profile data (Table 1) and parameters of soil water retention curves and hydraulic conductivity functions (Table 2).

**Table 1.** Soil physical parameters, horizon classification according to [34].

| Horizon | Depth from to (cm) | Sand (%) | Clay (%) | Silt (%) | Organic Carbon (%) | Bulk Density (g cm$^{-3}$) |
|---|---|---|---|---|---|---|
| | | | **Plot 1** | | | |
| Ap | 0–30 | 90 | 7 | 3 | 0.45 | 1.45 |
| Ael | 30–60 | 90 | 5 | 5 | 0.26 | 1.50 |
| Bt | 60–90 | 80 | 12 | 8 | 0.10 | 1.55 |
| C1 | 90–120 | 90 | 4 | 6 | - | - |
| C2 | 120–200 | 90 | 3 | 7 | - | - |
| | | | **Plot 2** | | | |
| Ap | 0–30 | 85 | 5 | 10 | 0.45 | 1.45 |
| Ael | 30–90 | 90 | 5 | 5 | 0.26 | 1.50 |
| Bt1 | 90–130 | 80 | 12 | 8 | 0.10 | 1.55 |
| Bt2 | 130–170 | 80 | 10 | 10 | - | - |
| C | 170–200 | 90 | 5 | 5 | - | - |
| | | | **Plot 3** | | | |
| Ap | 0–30 | 85 | 6 | 9 | 0.45 | 1.45 |
| Ael | 30–100 | 90 | 5 | 5 | 0.26 | 1.50 |
| Bt1 | 100–110 | 81 | 13 | 7 | 0.10 | 1.55 |
| Bt2 | 110–200 | 80 | 11 | 9 | - | - |

**Table 2.** Parameters $\theta_s$, $\theta_r$, n and $\alpha$ for $\theta(h)$-and $K(h)$-functions and saturated hydraulic conductivities $K_{sat}$, data from [35] (modified).

| | $\theta_s$ (cm$^3$ cm$^{-3}$) | $\theta_r$ (cm$^3$ cm$^{-3}$) | n | $\alpha$ (cm$^{-1}$) | $K_{sat}$ (cm day$^{-1}$) |
|---|---|---|---|---|---|
| | | **Plot 1** | | | |
| Ap | 0.38 | 0.03 | 2.013 | 0.021 | 92 |
| Ael | 0.32 | 0.03 | 2.179 | 0.027 | 162 |
| Bt | 0.38 | 0.07 | 2.147 | 0.028 | 30 |
| C1, C2 | 0.32 | 0.03 | 2.379 | 0.027 | 162 |
| | | **Plot 2** | | | |
| Ap | 0.39 | 0.03 | 2.013 | 0.021 | 92 |
| Ael | 0.32 | 0.03 | 2.179 | 0.027 | 162 |
| Bt1, Bt2 | 0.39 | 0.07 | 2.147 | 0.028 | 30 |
| C | 0.32 | 0.03 | 2.379 | 0.027 | 162 |

**Table 2.** *Cont.*

|  | $\theta_s$ (cm³ cm⁻³) | $\theta_r$ (cm³ cm⁻³) | n | $\alpha$ (cm⁻¹) | $K_{sat}$ (cm day⁻¹) |
|---|---|---|---|---|---|
| | | | **Plot 3** | | |
| Ap | 0.39 | 0.03 | 2.013 | 0.021 | 92 |
| Ael | 0.32 | 0.03 | 2.379 | 0.027 | 162 |
| Bt1, Bt2 | 0.39 | 0.07 | 2.147 | 0.028 | 30 |

The soil type is an Eutric cambisol according to the Food and Agriculture Organization FAO-classification [36]. This field experiment was established to investigate the influence of different cropping systems at each plot on crop and soil parameters and to generate datasets for agroecosystem modelling, model parametrization, model validation and model comparison. Three different farming systems with different intensity levels were investigated from 1993 to 1998 (Tables A1–A3). From 1999 to 2000, lucerne-clover-grass-mix was cultivated at all three plots to start another farming experiment (Tables A1–A3). After that period, a tillage experiment was established from 2001 up to 2006 to analyze the impact of different tillage systems on yield, crop growth, soil water and matter fluxes. From 2007 up to 2014, an experiment with two different crop rotations with bio-energy crops was established to investigate the long-term influence of such rotations on soil fertility. For all three plots, crop types, sowing and harvest data, and yield data are summarized in Tables A1–A3. More detailed information about this Experimental Station and the field experiments can be obtained from [37,38].

### 2.3. Soil Hydrological Measurements

At each of the three field plots, TRIME-(Time Domain Reflectometry with Intelligent MicroElements)-probes (Manufacturer IMKO, Germany) were installed vertically for an automatic measurement of daily volumetric depth-averaged soil water contents. The length of the probes at 26 cm (probe head: 10 cm, rod length: 16 cm) enabled the determination of soil water contents for soil compartments 0–30, 30–60, 60–90, 90–120 and 120–150 cm depth (Figure 1). This measurement setup was selected for the determination of total soil water storage in the root zone.

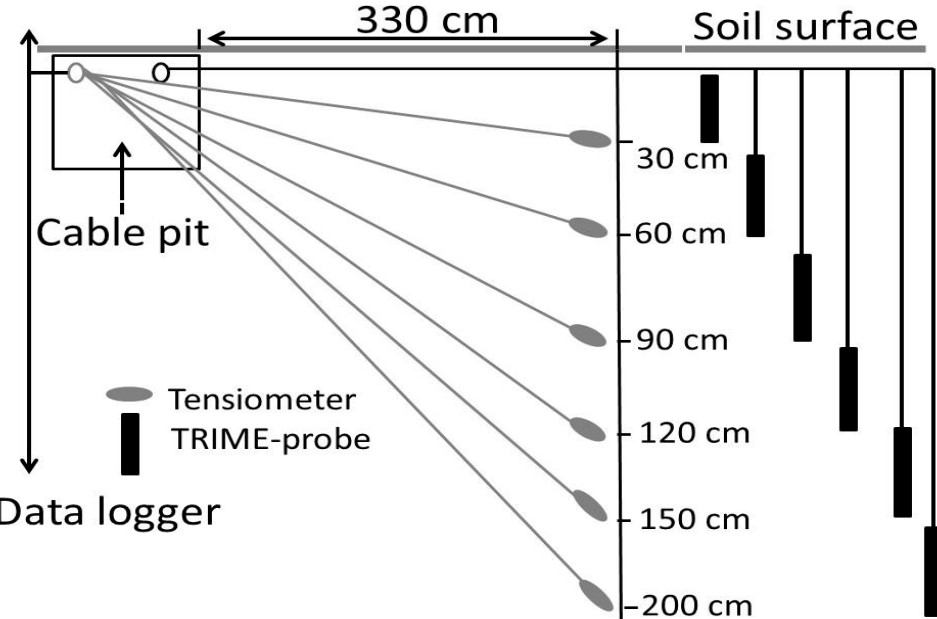

**Figure 1.** Installation of Time Domain Reflectometry with Intelligent MicroElements (TRIME) and tensiometer at the three field plots ([38], modified).

On each plot, soil samples were taken at 4–8 sampling times per year from the soil compartments 0–30, 30–60 and 60–90 cm depth for the determination of soil water contents using the gravimetric method in separate selected years. The measurements of the TRIME-probes were corrected using these gravimetrically determined soil water contents and the mean measurement error was estimated at ±0.025 cm$^3$ cm$^{-3}$ [38].

Respectively one tensiometer was installed at 30, 60, 90, 120 and 150 cm depth for the observation of daily pressure heads (Figure 1). During frost periods in the winter half years, tensiometers at 30 and 60 cm depth had to be removed to avoid frost damage. The TRIME-probes and the tensiometers were connected to a data logger. The measurements cover the time period from 1 January 1993 to 31 December 2014.

Due to the measurement setup, the TRIME-probes measure depth averaged soil water contents for a soil compartment with a thickness of app. 20–25 cm in a spherical soil volume with a diameter between 5 and 10 mm along the sensing rods of the TRIME-probes with an accuracy of ± 0.025 cm$^3$ cm$^{-3}$. Tensiometers determine pressure heads for a spherical soil volume with a diameter of 5–10 cm around the porous cups of the tensiometers with a precision of ±1–5 hPa [10].

The time series of observed soil water contents and pressure heads were checked by visual inspection for a detection of errors. Frozen soil conditions led to abrupt decreases of measured soil water contents because TRIME-probes detect only liquid water. These abrupt decreases ranged between 0.09 and 0.15 cm$^3$ cm$^{-3}$. Therefore, periods with frozen soil conditions and corresponding erroneous observed soil water contents were excluded from our analysis. After that, time series of measured soil water contents and pressure heads were checked using daily rainfall rates (precipitation→increase in soil moisture; no rainfall and evapotranspiration > 0→decrease in soil moisture). Finally, measured soil water contents and pressure heads were compared (increase of soil water contents→decrease of pressure heads; decrease of soil water contents→increase of pressure heads). Data which showed errors in both steps were flagged as erroneous and excluded from the analysis. This concerns about 15% of the measured data, and more details can be found in [38].

### 2.4. Estimation of Model Performance

In our study, we used the modelling efficiency index IA [39], coefficient of determination R$^2$ and the root-mean-square-deviation (RMSD).

$$
\begin{aligned}
\text{(a) } IA &= 1 - \frac{\sum_{i=1}^{n}(\theta_{sim}-\theta_{obs})^2}{\sum_{i=1}^{n}\left[\left|\theta_{sim}-\theta_{obs-mean}\right|+\left|\theta_{obs}-\theta_{obs-mean}\right|\right]^2} \\
\text{(b) } R^2 &= \left[\frac{\sum_{i=1}^{n}(\theta_{obs}-\theta_{obs-mean})-(\theta_{sim}-\theta_{sim-mean})}{\sqrt{\sum_{i=1}^{n}(\theta_{obs}-\theta_{obs-mean})^2}-\sqrt{\sum_{i=1}^{n}(\theta_{sim}-\theta_{sim-mean})^2}}\right]^2 \\
\text{(c) } RMSD &= \sqrt{\frac{\sum_{i=1}^{n}(\theta_{sim}-\theta_{obs})^2}{n}},
\end{aligned}
\tag{8}
$$

where $\theta_{sim}$ and $\theta_{obs}$ are simulated and observed values; n means the number of data pairs, and $\theta_{obs-mean}$ and $\theta_{im-mean}$ are mean values. IA and R$^2$ are in a range from 0 to 1. A value of 1 suggests a perfect fit of simulated to observed values.

### 2.5. Model Set Up

The model calculations were carried out with a daily time step from 1 January 1993 to 31 December 2014. The soil profiles were discretized into 20 computation layers, each having a thickness of 10 cm and the downward calculated soil water flux at 200 cm depth was treated as bottom flux. The lower boundary condition for the simulation of soil water fluxes was free drainage. The initial soil water contents for the model calculations were set to soil moisture values at pressure heads of 100 hPa, which corresponds approximately to field capacity (Table 2). Simulated soil water contents of the corresponding 10 cm layers were used for the calculation of mean values of the soil compartments at 0–30, 30–60, 60–90, 90–120 and 120–150 cm depth to enable a comparison with the TRIME-measurements.

For the determination of total soil water storage, simulated soil water contents of the 10 cm layers were summarized down to the lower soil profile boundary at 200 cm depth. The measured pressure heads were compared with those simulated for the corresponding computation layer.

Simulation of crop growth was initialized by seeding date and was stopped at harvest. For the parametrization of the crop growth model, we used the WOFOST parameters for Central European conditions [26]. For the crops cultivated at these three field plots (Tables A1–A3), the values of ext_coeff (Equation (2)) ranged from 0.33 for winter oil seed rape up to 0.75 for potatoes [26].

## 3. Results

*3.1. Hydrometeorological Conditions*

During the 22-year period, annual rainfall ranged from 371 mm in 2006 up to 747 mm in 2002. The long-term mean annual precipitation from 1951 to 2000 is at 565 mm.

In the hydrologic winter half years (November–April), positive values of climatic water balance due to rainfall surplus were calculated (Figure 2). Only in the winter half year 1995/1996, a climatic water deficit due to rainfall at 90 mm and $ET_p$ at 121 mm was calculated (Figure 2a).

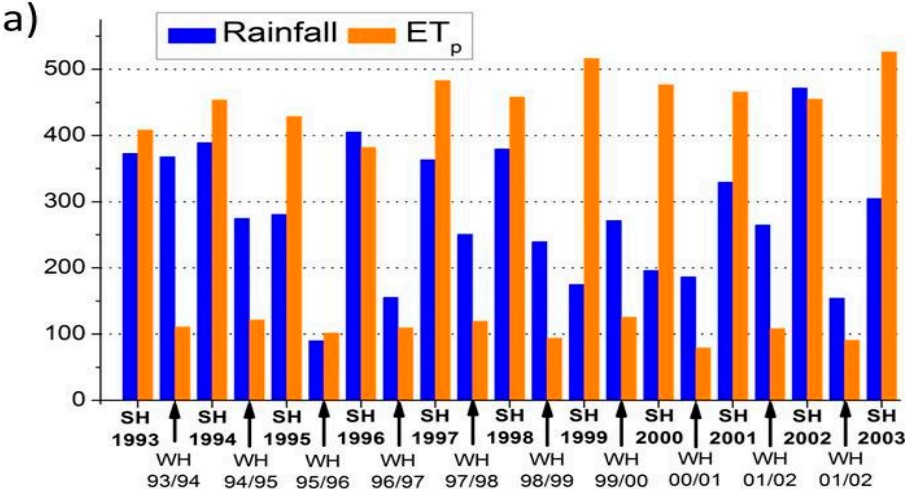

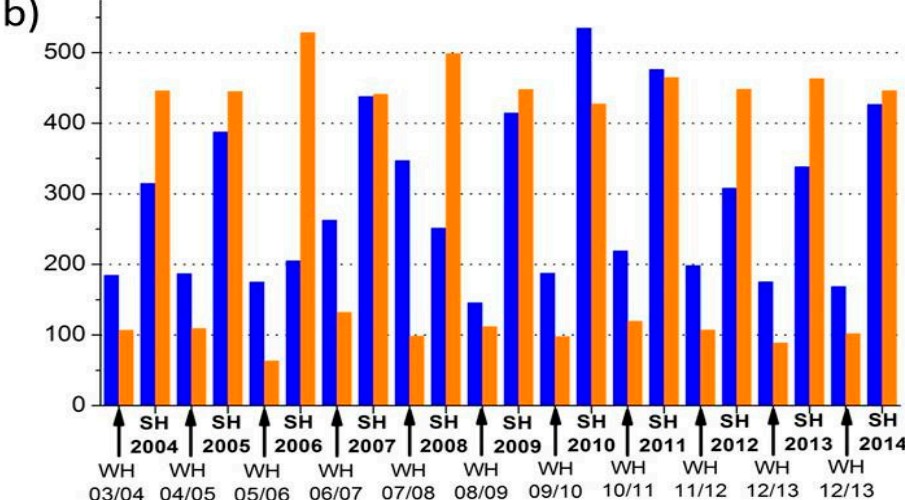

**Figure 2.** Rainfall and potential evapotranspiration ($ET_p$) for hydrological summer half years (SH) from May to October and for hydrological winter half years (WH) from November to April: (**a**), 1993–2003; (**b**) 2004–2014.

Climatic water surplus (= rainfall > $ET_p$) during the hydrologic summer half years (May–October) indicated wet hydrometeorological conditions in 1996 with rainfall of 406 mm, in 2002 with 480 mm (Figure 2a), in 2010 with 535 mm and in 2011 with 476 mm (Figure 2b).

The highest climatic water deficits (= rainfall < $ET_p$) in the summer half years indicating dry hydrometeorological conditions were observed in 1999 with −341 mm, in 2000 with −281 mm, in 2003 with −221 mm (Figure 2a) and in 2006 with −323 mm (Figure 2b).

### 3.2. Simulated LAI, Rooting Depth, RWU, Soil Water Storage and Drainage

Since we focused on soil hydrological conditions, we compared only simulated and observed yields for a rough estimation of the plausibility of the crop growth simulations. This comparison indicated plausible calculations of crop growth (Tables A1–A3). A more detailed analysis of the model performance for crop growth by comparing simulated with measured above-ground biomass at the three experimental field plots from 1993 to 1998 resulted in an IA from 0.90 to 0.99 [25].

Simulated daily values of LAI control the partitioning between RWU and soil evaporation whereas calculated rooting depth is the lower boundary of soil water withdrawal by RWU in the soil profiles (Equations (2)–(4), Figure 3). Thus, in addition to $ET_p$, the amount and temporal distribution of rainfall as well as soil water storage parameters, LAI and rooting depth have the highest impact on soil hydrological conditions in the summer half years. An impact of the different crop rotations on simulated RWU-rates was observed in the period from 2002 to 2012 (Figure 4). In this period, the different crops cultivated at the three plots showed higher variations in simulated LAI and rooting depth (Figures 3 and 4, Tables A1–A3).

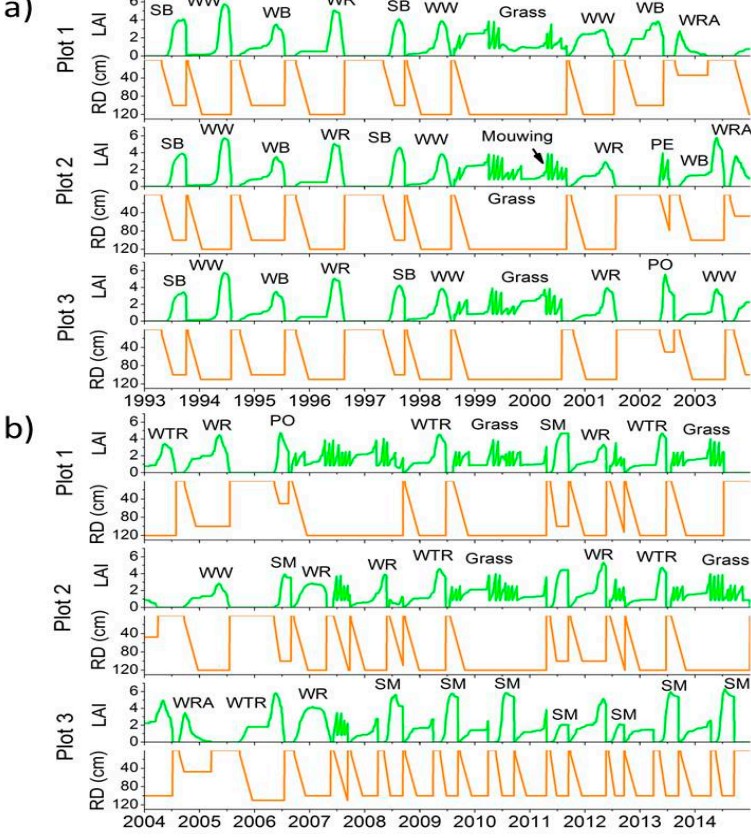

**Figure 3.** Simulated daily leaf area index (LAI) and rooting depth (RD), Plot 1–3 (SB = Sugar beet, WW = Winter wheat, WR = Winter rye, WB = Winter barley, Grass = Lucerne-clover-grass-mixture, WRA = Winter oil seed rape, PE = Peas, PO = Potatoes, WTR = Winter triticale, SM = Silage maize): (**a**) 1993–2003; (**b**) 2004–2014.

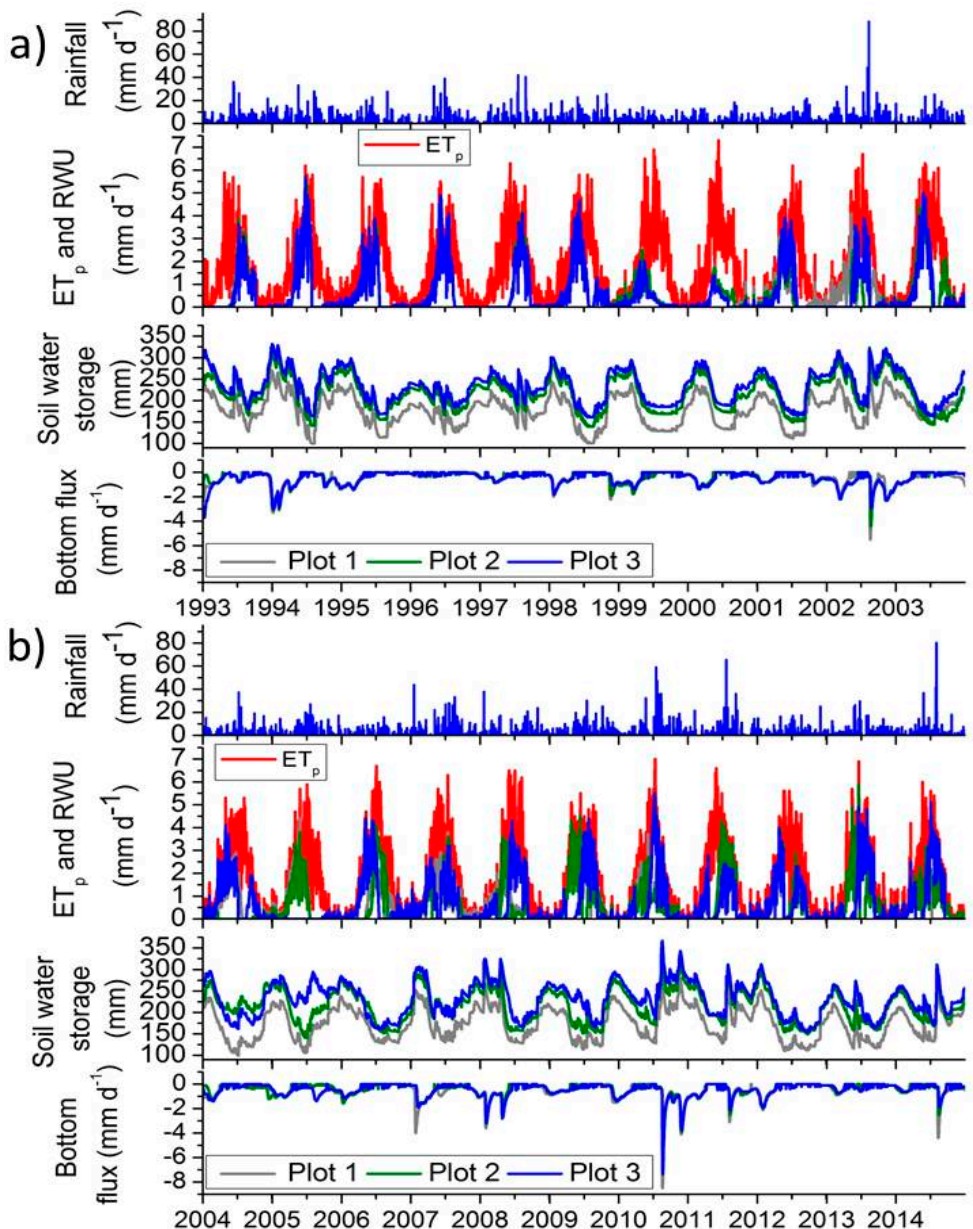

**Figure 4.** Daily rainfall, simulated $ET_p$ and actual root water uptake (RWU), soil water storage 0–200 cm depth and bottom flux at 200 cm depth, Plot 1–3: (**a**) 1993–2004; (**b**) 2004–2014.

The comparison of $ET_p$-rates with actual RWU-rates in the summer half years indicated the strong impact of the amount of soil water storage and corresponding soil water availability on RWU since RWU-rates were mainly significantly lower than $ET_p$, especially in the dry years 1999 and 2000 (Figures 2–4). Only in wet years such as 2010, RWU-rates were closer to $ET_p$-rates (Figures 2–4). In the winter half years, simulated daily soil water storage for the compartment 0–200 cm depth increased due to rainfall surplus and minor or no soil water withdrawal by RWU and evaporation. In the summer half years, soil water storage decreased due to soil water depletion by increased RWU and evaporation (Figure 4). However, the highest simulated soil water storage values between 297 and 364 mm were observed in the summer half years of 2002 and 2010 due to highest rainfall between 480 and 535 mm (Figures 2 and 4). Simulated values of daily soil water storage at Plot 2 and Plot 3 were higher due to higher field capacity and wilting point than those calculated at Plot 1 (Figure 4, Table 3).

**Table 3.** Porosity for 200 cm soil depth (PV_200 cm, 0 hPa), field capacity (FC_200 cm, 100 hPa), and wilting point (WP_200 cm, 15,000 hPa) calculated using vGM-data from Table 2.

|  | PV_200 cm (mm) | FC_200 cm (mm) | WP_200 cm (mm) |
|---|---|---|---|
| Plot 1 | 678 | 238 | 65 |
| Plot 2 | 710 | 266 | 81 |
| Plot 3 | 724 | 267 | 88 |

Longer drainage periods were calculated for the winter half years (Figure 4). However, the highest daily rainfall rates with 88 mm day$^{-1}$ at 12 August 2002, 59 mm day$^{-1}$ at 17 July 2010, 65 mm day$^{-1}$ at 22 July 2011 and with 80 mm day$^{-1}$ at 3 August 2014 resulted in the highest calculated drainage peaks from −4 to −8.8 mm day$^{-1}$ within the total simulation period (Figure 4). These drainage peaks at Plot 3 were lower than those calculated at Plot 1 and Plot 2 (Figure 4). This was attributed to lower K$_{sat}$ at 32 cm day$^{-1}$ at the lower boundary of the soil profile and higher field capacity at Plot 3 as compared to K$_{sat}$-values at 162 cm day$^{-1}$ and lower field capacity at Plot 1 and Plot 2 (Tables 2 and 3).

*3.3. Soil Water Contents and Pressure Heads at Plot 1*

Simulated soil water contents at the soil compartment 0–30 cm in the summer half years decreased to values at 0.03 cm$^3$ cm$^{-3}$ and were lower than those measured by TRIME with the lowest values at 0.07 cm$^3$ cm$^{-3}$ whereas simulated and measured soil water contents in the winter half years were in the same order of magnitude (Figures 5a and 6a). Due to a sand content of 90% in this soil compartment (Table 1), a higher wilting point between 0.07 and 0.10 cm$^3$ cm$^{-3}$ as reason for this mismatch between simulated and measured soil water contents was estimated as unlikely [34,35]. These differences between simulated and measured soil water contents up to 0.09 cm$^3$ cm$^{-3}$ in the summer half years were higher than the estimated measurement error of the TRIME-probes at ±0.025 cm$^3$ cm$^{-3}$. This suggested higher simulated soil water extraction by RWU and evaporation as compared to the TRIME-measurements. However, simulated soil water contents in the summer half years were similar to those determined by gravimetry and calculated and measured pressure heads at 30 cm depth run mainly similar between −20 and −880 hPa as well (Figure 5a,b and Figure 6a,b). This, in turn, indicated an adequate calculation of soil water depletion by RWU and evaporation. Thus, we attributed these mismatches between simulations and observations to erroneous TRIME-measurements, particularly from 2007 to 2014 (Figure 6a). In spite of these mismatches, the model performance for soil water contents at the 0–30 cm compartment was described by an IA at 0.82, R$^2$ at 0.61 and RMSD at 0.02 cm$^3$ cm$^{-3}$ (Table 4). The comparison of simulated with measured pressure heads at 30 cm depth resulted in an IA at 0.78, R$^2$ at 0.50 and RMSD at 120 hPa (Table 5).

At the compartment 30–60 cm, simulated and measured soil water contents were mainly in the same order of magnitude between 0.02 and 0.18 cm$^3$ cm$^{-3}$ (Figures 5c and 6c). Thus, IA was at 0.84, R$^2$ at 0.63 and RMSD at 0.02 cm$^3$ cm$^{-3}$ (Table 4). Only from 1995 to 1996, soil water contents measured by TRIME ranged above the simulated ones (Figure 5c). However, pressure heads observed at 60 cm depth declined down to values between −850 and −900 hPa in the summer half years of 1996, 1997 and 2003 whereas corresponding simulated pressure heads showed only minor decreases (Figure 5d). Therefore, IA was only at 0.51, R$^2$ at 0.20 and RMSD at 140 hPa (Table 5). This suggested an underestimation of RWU by the model in this part of the root zone for these three summer half years. However, this underestimation was confirmed by the observed soil water contents only in 2003 (Figure 5c). In this dry and hot year, measured pressure heads at 60 cm depth and soil water contents observed at the soil compartment 30–60 cm suggested higher soil water depletion by RWU as compared to the model calculations (Figure 5c,d). However, the goodness of fit between simulated soil water contents and those measured by TRIME and gravimetry in the other summer half years indicated adequate calculations of RWU by the model (Figures 5c and 6c). Thus, these mismatches between measurements and simulations were attributed to erroneous TRIME-measurements in 1995 and 1996 as well as to an underestimation of RWU by the model in the summer half years of 1997 and 2003.

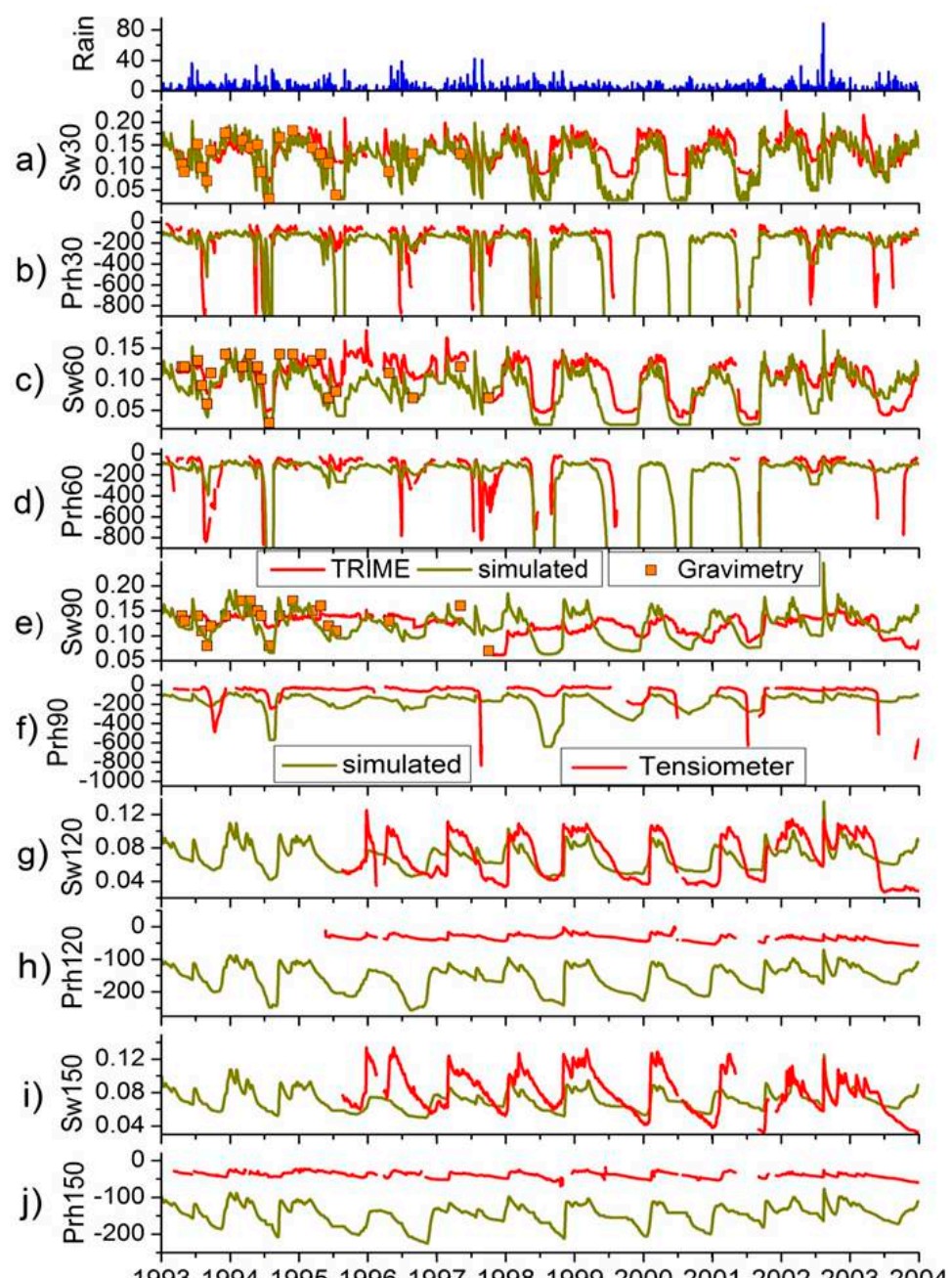

**Figure 5.** Daily rainfall (Rain) in mm d$^{-1}$, simulated and measured soil water contents (TRIME, Gravimetry) in cm$^3$ cm$^{-3}$ at soil compartments 0–30 cm (**a**, Sw30), 30–60 cm (**c**, Sw60), 60–90 cm (**e**, Sw90), 90–120 cm (**g**, Sw120), and 12–150 cm depth (**i**, Sw150) and pressure heads in hPa at 30 cm (**b**, Prh30), 60 cm (**d**, Prh60), 90 cm (**f**, Prh90), 120 cm (**h**, Prh120) and 150 cm depth (**j**, Prh150), Plot 1, 1993–2003.

At the soil compartment 60–90 cm, simulated soil water contents in most of the summer half years were distinctly lower than those measured by TRIME (Figures 5e and 6e). In addition, simulated pressure heads at 90 cm depth in the summer half years of 1994, 1998, 2004, 2007 and 2009 showed higher decreases than the measured ones (Figures 5f and 6f). This suggested an overestimation of RWU by the model. Vice versa, soil water depletion by RWU indicated by declining measured pressure heads and low observed soil water contents in the summer half year of 2003 was not simulated adequately by the model (Figure 5e,f).

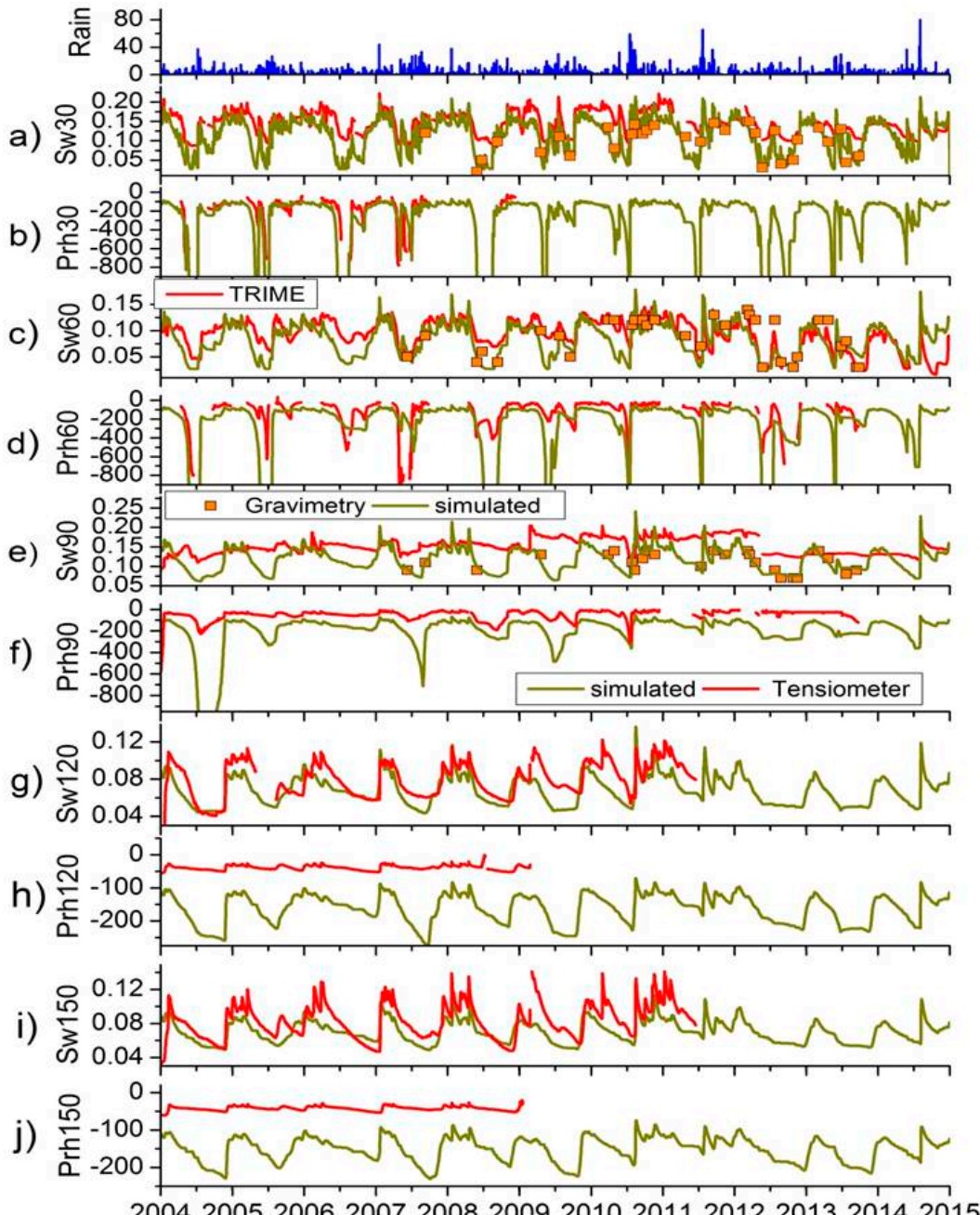

**Figure 6.** Daily rainfall (Rain) in mm d$^{-1}$, simulated and measured soil water contents (TRIME, Gravimetry) in cm$^3$ cm$^{-3}$ at soil compartments 0–30 cm (**a**, Sw30), 30–60 cm (**c**, Sw60), 60–90 cm (**e**, Sw90), 90–120 cm (**g**, Sw120), and 120–150 cm depth (**i**, Sw150) and pressure heads in hPa at 30 cm (**b**, Prh30), 60 cm (**d**, Prh60), 90 cm (**f**, Prh90), 120 cm (**h**, Prh120) and 150 cm depth (**j**, Prh150), Plot 1, 2004–2014.

However, soil water contents determined by gravimetry in the summer half years showed a good fit to the simulated ones, particularly in 1994 and from 2007 to 2014 (Figures 5e and 6e). This, in turn, indicated adequate RWU-calculations by the model. Therefore, we attributed these differences between simulated and measured soil water contents to an underestimation of RWU by the model in 2003 and to erroneous TRIME-measurements, especially from 2007 to 2014. Such erroneous measurements might be caused by soil compaction around the rods of the probes, deformations of the rods by stones during the installation, or electronic failures in the TRIME-probes, e.g., [40,41]. These mismatches

between simulated and measured pressure heads and soil water contents resulted in an IA from 0.30 to 0.45, $R^2$ at 0.10 and RMSD at 0.04 cm$^3$ cm$^{-3}$ and at 200 hPa (Tables 4 and 5).

**Table 4.** Modelling performance for soil water contents.

| Soil Compartment | Number of Data Pairs (1993–2014 = 8035 Potential Data Pairs Measured Versus Simulated Daily Soil Water Content) | IA | $R^2$ | RMSD (cm$^3$ cm$^{-3}$) |
|---|---|---|---|---|
| **Plot 1** | | | | |
| 0–30 cm | 6900 | 0.82 | 0.61 | 0.02 |
| 30–60 cm | 7409 | 0.84 | 0.63 | 0.02 |
| 60–90 cm | 7667 | 0.45 | 0.10 | 0.04 |
| 90–120 cm | 5565 | 0.78 | 0.55 | 0.02 |
| 120–150 cm | 5551 | 0.72 | 0.50 | 0.02 |
| **Plot 2** | | | | |
| 0–30 cm | 6720 | 0.78 | 0.58 | 0.03 |
| 30–60 cm | 6658 | 0.83 | 0.62 | 0.02 |
| 60–90 cm | 6724 | 0.71 | 0.44 | 0.04 |
| 90–120 cm | 5915 | 0.52 | 0.12 | 0.04 |
| 120–150 cm | 6778 | 0.42 | 0.12 | 0.04 |
| **Plot 3** | | | | |
| 0–30 cm | 6720 | 0.86 | 0.66 | 0.02 |
| 30–60 cm | 7101 | 0.84 | 0.63 | 0.02 |
| 60–90 cm | 7221 | 0.82 | 0.61 | 0.02 |
| 90–120 cm | 5890 | 0.82 | 0.65 | 0.02 |
| 120–150 cm | 5590 | 0.71 | 0.53 | 0.02 |

**Table 5.** Modelling performance for pressure heads.

| Measurement Depth | Number of Data Pairs (1993–2014 = 8035 Potential Data Pairs Measured Versus Simulated Daily Pressure Head) | IA | $R^2$ | RMSD (hPa) |
|---|---|---|---|---|
| **Plot 1** | | | | |
| 30 cm | 2569 | 0.78 | 0.50 | 120 |
| 60 cm | 3985 | 0.51 | 0.20 | 140 |
| 90 cm | 6408 | 0.30 | 0.10 | 200 |
| 120 cm | 4756 | 0.11 | 0.11 | 122 |
| 150 cm | 5437 | 0.11 | 0.11 | 110 |
| **Plot 2** | | | | |
| 30 cm | 2592 | 0.79 | 0.56 | 141 |
| 60 cm | 2767 | 0.45 | 0.18 | 144 |
| 90 cm | 6938 | 0.33 | 0.10 | 102 |
| 120 cm | 5356 | 0.67 | 0.21 | 31 |
| 150 cm | 6137 | 0.78 | 0.67 | 20 |
| **Plot 3** | | | | |
| 30 cm | 2276 | 0.77 | 0.48 | 121 |
| 60 cm | 2554 | 0.28 | 0.11 | 136 |
| 90 cm | 6663 | 0.35 | 0.11 | 125 |
| 120 cm | 4826 | 0.74 | 0.45 | 27 |
| 150 cm | 5322 | 0.76 | 0.53 | 17 |

At the compartments 90–120 and 120–150 cm, measured soil water contents in the winter half years ranged above the simulated ones, whereas in the summer half years observed and simulated soil water contents were mainly similar (Figure 5g,i and Figure 6g,i). The decrease of measured and simulated

soil water contents down to 0.03 cm$^3$ cm$^{-3}$ in most of the summer half years suggested soil water withdrawal by RWU in this part of the soil profile. Only in the dry summer half year of 2003, measured soil water contents down to 0.03 cm$^3$ cm$^{-3}$ were lower than the simulated ones with the lowest values at 0.06 cm$^3$ cm$^{-3}$ (Figure 5g,i). This indicated higher RWU as compared to the model calculations. Obviously, winter oil seed rape cultivated from August 2002 to July 2003 (Figure 3a, Table A1) showed an increased RWU in these deeper soil layers to compensate soil water stress conditions in the upper soil layers. This compensation was not simulated by the model (Figure 5g,i) due to a calculated maximum rooting depth of winter oilseed rape only at 40 cm (Figure 3a). Despite these mismatches, IA ranged between 0.78 and 0.72, R$^2$ from 0.55 to 0.50 and RMSD at 0.02 cm$^3$ cm$^{-3}$ (Table 4).

Measured pressure heads at 120 cm and 150 cm depth between −15 and −65 hPa suggested soil water contents near or above field capacity without any soil withdrawal by RWU (Figure 5h,j and Figure 6h,j). This was in contradiction to the soil water contents between 0.03 and 0.14 cm$^3$ cm$^{-3}$ measured by TRIME at the soil compartments 90–120 cm and 120–150 cm (Figure 5g,i and Figure 6g,i).

These contradictions between measured soil water contents and observed pressure heads were attributed to the previous mentioned different measurement volumes, precision and measuring principles (see Section 2.3). Simulated pressure heads, however, were from −90 to −250 hPa (Figure 5h,j and Figure 6h,j). These differences led to an IA and R$^2$ at 0.11 and RMSD between 110 and 122 hPa (Table 5).

### 3.4. Soil Water Contents and Pressure Heads at Plot 2

Simulated soil water contents at the soil compartment 0–30 cm in the summer half years of 1994, 1995, 1996, 1998, 2002, 2003, 2005 and from 2008 to 2014 decreased to values at 0.03 cm$^3$ cm$^{-3}$ and were lower than the measured ones between 0.06 and 0.12 cm$^3$ cm$^{-3}$ (Figures 7a and 8a). Despite these mismatches, IA was at 0.78, R$^2$ at 0.58 and RMSD at 0.03 cm$^3$ cm$^{-3}$ (Table 4). At the soil compartment 30–60 cm, simulated soil water contents in the summer half years of 1994, 1995, 1996, 1998, 2002, 2003 and 2005 down to 0.03 cm$^3$ cm$^{-3}$ were also lower than those measured by TRIME between 0.06 and 0.10 cm$^3$ cm$^{-3}$ (Figures 7c and 8c). In contrast to the soil compartment 0–30 cm, simulated and measured soil water contents at the 30–60 cm soil compartment showed a good match from 2008 to 2014 (Figure 8c). Thus, IA was at 0.83, R$^2$ at 0.62 and RMSD was at 0.02 cm$^3$ cm$^{-3}$ (Table 4).

In addition, measured pressure heads at 30 and 60 cm depth in these summer half years showed lower decreases as compared to the simulated ones with distinct higher declines (Figure 7b,d and Figure 8b,d). This suggested an overestimation of soil withdrawal by evaporation and RWU by the model for these upper two soil compartments. This can be caused by

- Inadequate description of the spatial distribution of RWU in the root zone depending on crop specific maximum rooting depth and root density distribution.
- Overestimation of potential RWU by the model.

Potential RWU is determined by the amount of calculated ET$_P$ and actual LAI, which depends on the current phenological development stage of the crop.

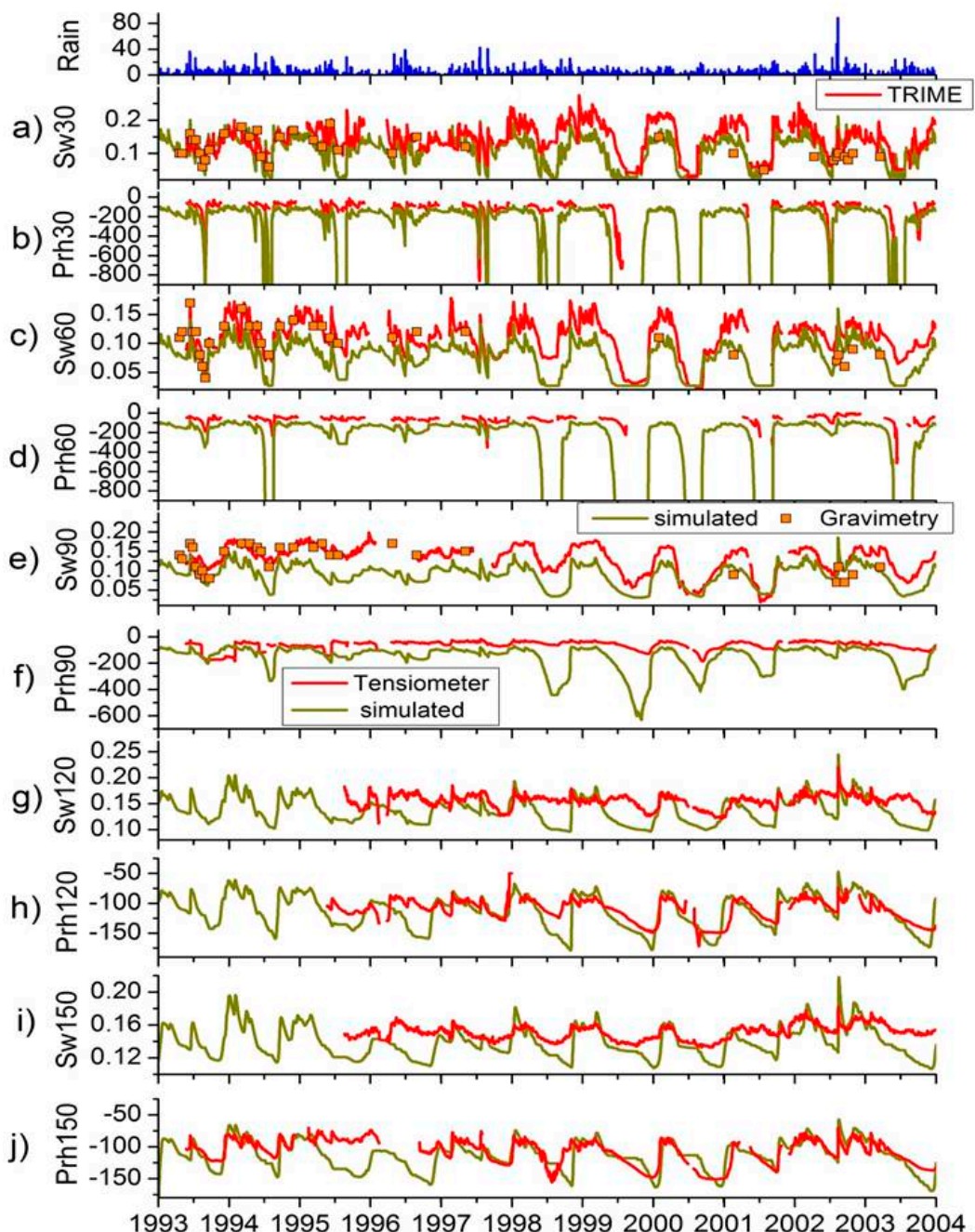

**Figure 7.** Daily rainfall (Rain) in mm d$^{-1}$, simulated and measured soil water contents (TRIME, Gravimetry) in cm$^3$ cm$^{-3}$ at soil compartments 0–30 cm (**a**, Sw30), 30–60 cm (**c**, Sw60), 60–90 cm (**e**, Sw90), 90–120 cm (**g**, Sw120) and 120–150 cm depth (**i**, Sw150) and pressure heads in hPa at 30 cm (**b**, Prh30), 60 cm (**d**, Prh60), 90 cm (**f**, Prh90), 120 cm (**h**, Prh120) and 150 cm depth (**j**, Prh150), Plot 2, 1993–2003.

Due to higher simulated soil water withdrawal in these summer half years, calculated soil water contents up to 0.20 cm$^3$ cm$^{-3}$ in the consecutive winter half years such as 1994/1995, 1995/1996 and 1998/1999 were lower than the measured ones with values up to 0.26 cm$^3$ cm$^{-3}$ (Figure 7a,c). From 1999 to 2002 and from 2006 to 2007, simulated and measured soil water contents showed mostly a good fit for both soil compartments (Figures 7a,c and 8a,c).

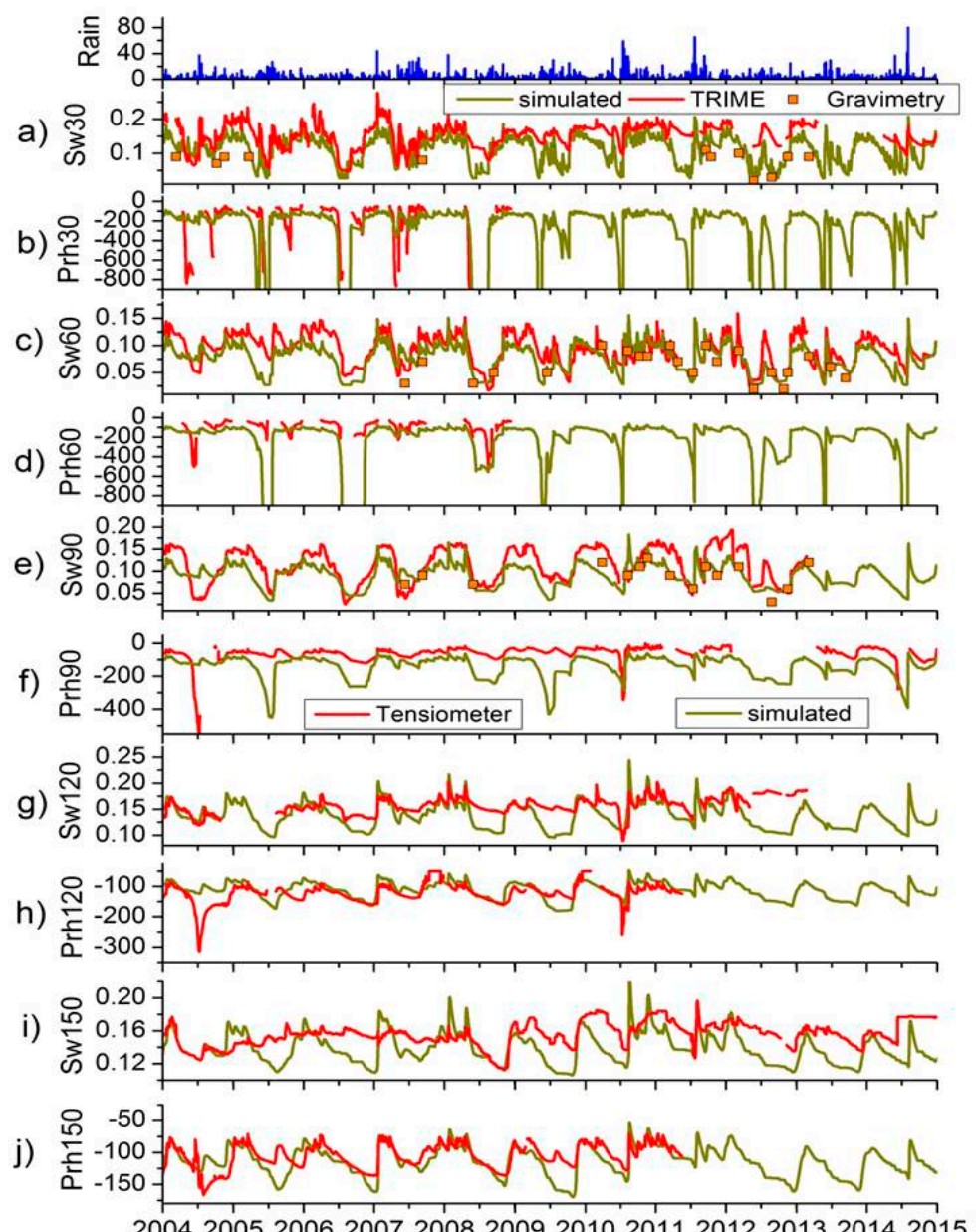

**Figure 8.** Daily rainfall (Rain) in mm d$^{-1}$, simulated and measured soil water contents (TRIME, Gravimetry) in cm$^3$ cm$^{-3}$ at soil compartments 0–30 cm (**a**, Sw30), 30–60 cm (**c**, Sw60), 60–90 cm (**e**, Sw90), 90–120 cm (**g**, Sw120), and 120–150 cm depth (**i**, Sw150) and pressure heads in hPa at 30 cm (**b**, Prh30), 60 cm (**d**, Prh60), 90 cm (**f**, Prh90), 120 cm (**h**, Prh120) and 150 cm depth (**j**, Prh150), Plot 2, 2004–2014.

However, since soil water contents measured by gravimetry at the 0–30 cm compartment showed a good fit to the simulated ones in the summer half years, particularly from 2008 to 2014 (Figure 8a), we attributed these mismatches between simulated and measured soil water contents to erroneous TRIME-measurements in this period. Despite the previous mentioned differences between simulated and measured pressure heads in some summer half years, calculated and observed pressure heads at 30 cm depth showed a sufficient match suggested by an IA at 0.79, R$^2$ at 0.56 and RMSD at 141 hPa, whereas the comparison of simulated with measured pressure heads at 60 cm depth resulted in an IA at 0.45, R$^2$ at 0.18 and RMSD at 144 hPa (Figures 7b,d and 8b,d, Table 5).

At the soil compartment 60–90 cm, soil water contents measured by TRIME and gravimetry were up to 0.08 cm$^3$ cm$^{-3}$ higher than the simulated ones from 1994 to 1999 (Figure 7e). In the remaining

period from 2000 to 2014, simulated and measured soil water contents showed a better match, especially in the summer half years from 2001 to 2010. However, measured soil water contents in the winter half years were still higher than the simulated ones (Figures 7e and 8e). Thus, the model performance for soil water contents was described by an IA at 0.71, $R^2$ at 0.44 and RMSD at 0.04 cm$^3$ cm$^{-3}$ (Table 4).

Simulated pressure heads at 90 cm depth ranged between −75 and −650 hPa whereas measured pressure heads were from −10 to −120 hPa (Figures 7f and 8f). Therefore, the model performance was low with an IA at 0.33, $R^2$ at 0.10 and RMSD at 102 hPa (Table 5). This range of measured pressure heads suggested soil water contents near or above field capacity for the total investigation period. Therefore, the soil water contents at the compartment 60–90 cm measured by TRIME between 0.02 and 0.19 cm$^3$ cm$^{-3}$ were in contradiction to the observed pressure heads. Due to these contradictions, an identification of a reason for the mismatch between simulated and observed soil water contents from 1994 to 1999 and in the winter half years was hampered. Similar to Plot 1, these contradictions between measured soil water contents and observed pressure heads were attributed to the previous mentioned technical differences between TRIME-probes and tensiometers (see Section 2.3).

At the soil compartments 90–120 and 120–150 cm, simulated soil water contents in most of the summer half years decreased to values down to 0.10 cm$^3$ cm$^{-3}$ whereas corresponding measured soil water contents declined only to values between 0.13 and 0.15 cm$^3$ cm$^{-3}$ (Figures 7g,i and 8g,i). This resulted in an IA from 0.42 to 0.52, $R^2$ at 0.12 and RMSD at 0.04 cm$^3$ cm$^{-3}$ (Table 4) and suggested higher simulated soil water withdrawal by RWU in this lower part of the soil profile as compared to the observations. This calculated soil water withdrawal was due to simulated maximum rooting depths of the crops at 120 cm depth in these summer half years (Figure 3). Simulated and measured pressure heads at 120 and 150 cm depth were in the same range from −50 to −180 hPa (Figures 7 and 8). This resulted in an IA from 0.67 to 0.78, $R^2$ between 0.21 and 0.67 and RMSD from 20 to 31 hPa (Table 5). Despite the sufficient fit between simulated and measured pressure heads, we finally attributed the mismatches between simulated and observed soil water contents to an overestimation of RWU by the model in the summer half years in this part of the soil profile from 90 to 150 cm depth.

### 3.5. Soil Water Contents and Pressure Heads at Plot 3

At the soil compartment 0–30 cm, simulated soil water contents and those measured by TRIME and gravimetry showed a good fit from 1996 to 2009 (Figures 9a and 10a). However, from 1993 to 1995 and from 2010 to 2014, soil water contents measured by TRIME in the summer half years decreased to values at 0.07 cm$^3$ cm$^{-3}$ whereas simulated soil water contents declined to lower values at 0.03 cm$^3$ cm$^{-3}$ (Figures 9a and 10a). This indicated higher simulated soil water extraction by RWU and evaporation in the summer half years of these two periods as compared to the measurements, especially in 1995, 2011 and 2012 (Figures 9a and 10a). In contrast to that, the good fit of simulated soil water contents at the 0–30 cm soil compartment to those determined by gravimetry suggested an adequate calculation of RWU by the model, particularly from 2009 to 2014 (Figure 10a). Thus, we attributed these discrepancies between observations and simulations from 1993 to 1995 and from 2010 to 2014 to erroneous TRIME-measurements (Figures 9a and 10a). Despite these mismatches, IA for soil water contents was at 0.86, $R^2$ at 0.66 and RMSD at 0.02 cm$^3$ cm$^{-3}$ (Table 4). Measured and simulated pressure heads at 30 cm depth in the summer half years were mainly in the same order of magnitude (Figure 10b). Thus, IA was at 0.77, $R^2$ at 0.48 and RMSD at 121 hPa (Table 5).

From 1993 to 1999, soil water contents at the soil compartment 30–60 cm depth measured by TRIME and gravimetry were higher than the simulated ones (Figure 9c). In the following period from 2000 to 2014, observed and simulated soil water contents showed a distinct better match.

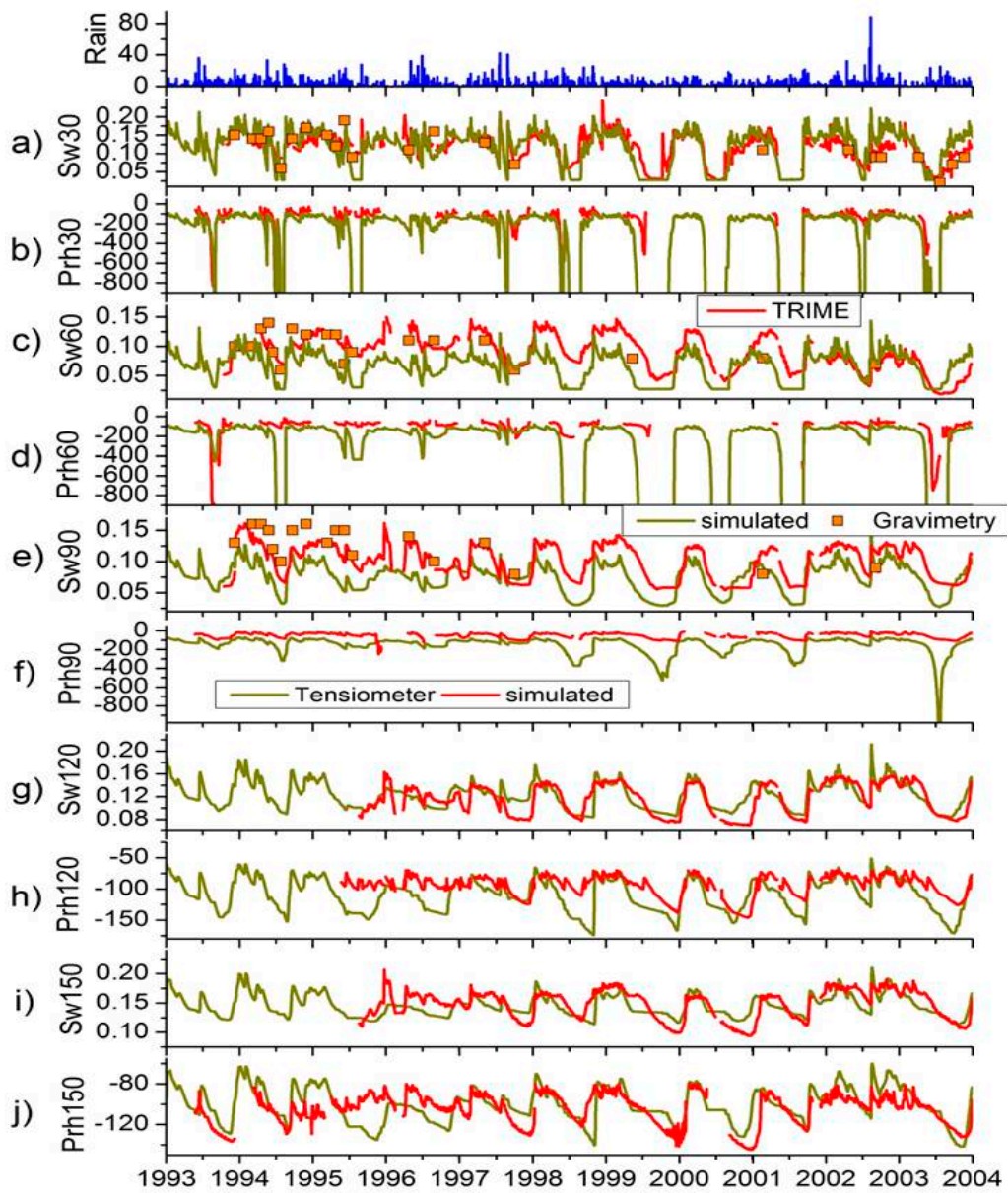

**Figure 9.** Daily rainfall (Rain) in mm d$^{-1}$, simulated and measured soil water contents (TRIME, Gravimetry) in cm$^3$cm$^{-3}$ at soil compartments 0–30 cm (**a**, Sw30), 30–60 cm (**c**, Sw60), 60–90 cm (**e**, Sw90), 90–120 cm (**g**, Sw120), and 120–150 cm depth (**i**, Sw150) and pressure heads in hPa at 30 cm (**b**, Prh30), 60 cm (**d**, Prh60), 90 cm (**f**, Prh90), 120 cm (**h**, Prh120) and 150 cm depth (**j**, Prh150), Plot 3, 1993–2003.

Therefore, IA was at 0.84, R$^2$ at 0.63 and RMSD at 0.02 cm$^3$ cm$^{-3}$ (Figures 9c and 10c, Table 4). Measured pressure heads at 60 cm depth showed no decreases suggesting no or only minor soil water depletion whereas simulated declining pressure heads indicated higher soil water withdrawal by RWU (Figures 9d and 10d). Due to these differences, IA was at 0.28, R$^2$ at 0.11 and RMSD at 136 hPa (Table 5). However, an explicit identification of reasons for this mismatch between measurements and simulations from 1993 to 1999 such as an overestimation of RWU by the model or erroneous TRIME-measurements was hampered since the time course of corresponding pressure heads measured at 60 cm depth showed a high amount of missing data (Figures 9d and 10d, Table 5).

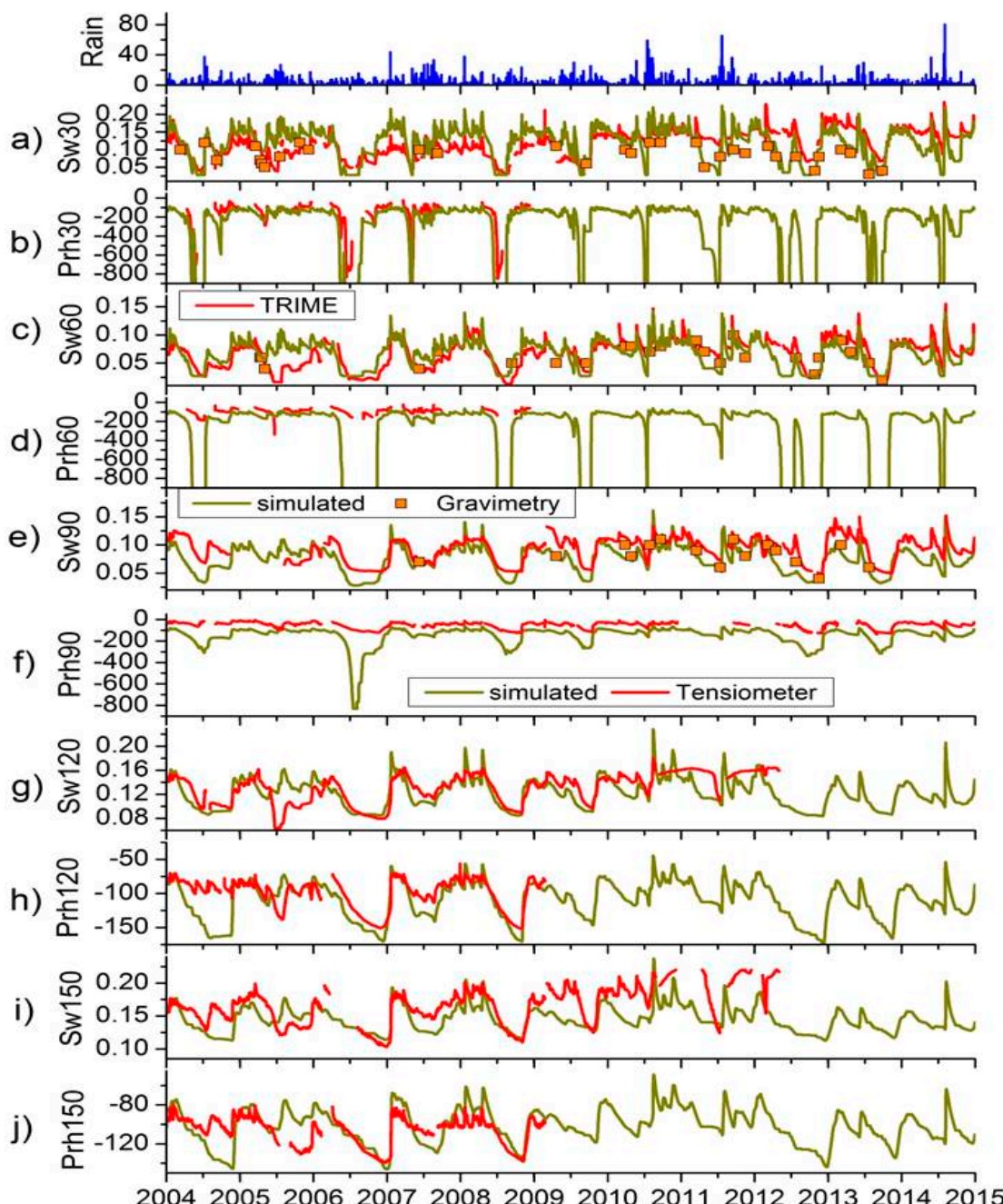

**Figure 10.** Daily rainfall (Rain) in mm $d^{-1}$, simulated and measured soil water contents (TRIME, Gravimetry) in $cm^3$ $cm^{-3}$ at soil compartments 0–30 cm (**a**, Sw30), 30–60 cm (**c**, Sw60), 60–90 cm (**e**, Sw90), 90–120 cm (**g**, Sw120), and 120–150 cm depth (**i**, Sw150) and pressure heads in hPa at 30 cm (**b**, Prh30), 60 cm (**d**, Prh60), 90 cm (**f**, Prh90), 120 cm (**h**, Prh120) and 150 cm depth (**j, Prh150**), Plot 3, 2004–2014.

From 1993 to 2000, measured soil water contents at the soil compartment 60–90 cm were higher than the simulated ones, similar to the soil compartment 30–60 cm (Figure 9c,e). In the summer half years of 2003, 2004 and 2006, simulated soil water contents decreased to values at 0.03 $cm^3$ $cm^{-3}$, which were lower than those measured by TRIME with the lowest values at 0.07 $cm^3$ $cm^{-3}$ (Figures 9e and 10e). This indicated higher simulated soil withdrawal by RWU as compared to the TRIME- measurements. However, from 2007 to 2014, simulated soil water contents and those measured by TRIME and gravimetry were mainly in the same order of magnitude (Figures 9e and 10e). Therefore, IA was at

0.82, $R^2$ at 0.61 and RMSD at 0.02 $cm^3$ $cm^{-3}$ (Table 4). Furthermore, measured pressure heads at 90 cm depth showed only minor decreases in these summer half years as compared to the simulated ones with distinct higher declines, particularly in the dry years 2003 and 2006 (Figures 9f and 10f). Thus, IA was at 0.35, $R^2$ at 0.11 and RMSD at 125 hPa (Table 5). Therefore, we attributed the differences between observations and simulations from 1993 to 2000 to erroneous TRIME-measurements and the discrepancies between simulated and measured soil water contents and pressure heads in the summer half years of 2003 and 2006 to an overestimation of RWU by the model.

Measured soil water contents at the soil compartment 90–120 cm in the summer half years 1997, 2000 and 2005 decreased to values at 0.07 $cm^3$ $cm^{-3}$, which were lower than the simulated ones with the lowest values at 0.09 $cm^3$ $cm^{-3}$ (Figure 9g). This suggested higher soil withdrawal by RWU in these summer half years as compared to the model calculations. In the other years, simulated soil and measured water contents were in the same order of magnitude (Figures 9g and 10g). Thus, the model performance was described by an IA at 0.82, $R^2$ at 0.65 and RMSD at 0.02 $cm^3$ $cm^{-3}$ (Table 4). Simulated pressure heads at 120 cm depth ranged below the measured ones only in the summer half years of 1998, 2003 and 2004. In the other years, simulated and measured pressure heads were similar (Figure 10h). Therefore, IA was at 0.74, $R^2$ at 0.45 and RMSD at 27 hPa (Table 5).

Simulated and measured soil water contents at the soil compartment 120–150 cm were mainly similar (Figures 9i and 10i). Only in the summer half years of 1997, 1999, 2000 and 2007, measured soil water contents were lower than the simulated ones, but the differences between simulated and observed soil water contents, were below the measurement error of the TRIME-probes at ±0.025 $cm^3$ $cm^{-3}$. Thus, IA was at 0.71, $R^2$ at 0.53 and RMSD at 0.02 $cm^3$ $cm^{-3}$ (Table 4). Calculated and observed pressure heads at 150 cm depth were also similar and, therefore, IA was at 0.76, $R^2$ at 0.52 and RMSD at 17 hPa (Table 5).

## 4. Discussion

Temporal simulated and measured soil water dynamics at all three plots were similar with increasing soil water storage during the winter half years due to rainfall surplus and decreasing soil water storage due to soil water withdrawal by evaporation and RWU in the summer half years (Figures 4–10). Differences in simulated drainage and soil water storage between the three experimental field plots (Figure 4) were attributed to the different depths of the soil horizons (Table 1), which caused differences in soil water storage parameters and hydraulic conductivity, particularly at the lower boundary of the soil profiles at 200 cm depth (Tables 2 and 3).

A distinct impact of the different crop rotations established at the three field plots on simulated soil water contents and pressure heads was observed in 2003, 2004, 2005 and 2009 (Figures 4–10, Tables A1–A3). In 2003, winter soil seed rape showed a simulated maximum rooting depth of 40 cm at Plot 1 in contrast to winter barley at Plot 2 and winter wheat at Plot 3 with simulated maximum rooting depths between 90 and 110 cm (Figure 3a, Tables A1–A3). In 2004, winter triticale at Plot 1 and winter barley at Plot 3 showed simulated maximum rooting depths from 120 to 90 cm whereas winter oil seed rape at Plot 2 showed a simulated maximum rooting depth at 45 cm. In 2005, the crop growth model simulated rooting depths between 90 and 120 cm for winter rye at Plot 1 and winter wheat at Plot 2 in contrast to Plot 3 again with winter oil seed rape with a calculated rooting depth at 45 cm. In 2009, lucerne-grass-cover at Plot 1 and 2 showed longer vegetation periods as compared to silage maize at Plot 3 (Figure 3b). This resulted in different amounts of calculated soil water withdrawal by RWU and, thus, to different simulated soil water contents and pressure heads from soil surface down to 120 cm depth in the summer half years (Figures 4–10). However, these simulated differences between the experimental field plots were only partly confirmed by the measurements.

The model performance for soil water contents in terms of IA ranged between 0.45 and 0.86, $R^2$ was from 0.10 to 0.66 and RMSD between 0.02 and 0.04 $cm^3$ $cm^{-3}$ (Table 4). At all three field plots, the best model performance for soil water contents was observed for the upper two soil compartments 0–30 and 30–60 cm with IA from 0.78 to 0.86, $R^2$ between 0.58 and 0.66 and RMSD at 0.02 $cm^3$ $cm^{-3}$

(Table 4). In other agricultural experimental field studies, the comparison of simulated soil water contents with those measured by TDR resulted in IA-values between 0.34 and 0.96 and $R^2$ from 0.27 to 0.91, e.g., [16–24]. The model performance for pressure heads in terms of IA ranged from 0.11 up to 0.80, $R^2$ was from 0.10 to 0.56 and RMSD between 17 and 200 hPa (Table 5). At all three plots, the best match between simulated and observed pressure heads was observed at 30 cm depth with IA from 0.77 to 0.79, $R^2$ between 0.48 and 0.56 and RMSD from 17 to 200 hPa (Table 5). In other studies, IA for pressure heads was between 0.54 and 0.93, e.g., [18].

Large differences in the model performance for soil water pressure heads at 120 and 150 cm depth were observed between Plot 1 with an IA at 0.11 and Plot 2 and 3 with IA-values between 0.67 and 0.78 (Table 5, Figures 5–10). At Plot 1, tensiometers installed at 120 cm and 150 cm depth are located in the C-horizon with a sand content at 90%. At Plot 2 and 3, corresponding tensiometers are located in the Bt-Horizons with sand contents at 80% (Figure 1, Table 1). These differences between Plot 1 and Plots 2 and 3 regarding soil texture led to different vGM-parameter sets in this lower part of the soil profile with an impact on simulated soil water storage and drainage (Figure 4). However, these differences between Plot 1 and Plots 2 and 3 regarding soil texture and soil hydraulic parameters showed no impact on the model performance of soil water contents measured at 90–120 and 120–150 cm compartment with IA-values between 0.42 and 0.82 (Table 4, Figures 5–10). Thus, these differences regarding soil texture and vGM-parameter sets between Plot 1 and Plots 2 and 3 as reason for the low model performance for soil water pressure heads at 120 and 150 cm depth at Plot 1 as compared to Plots 2 and 3 were estimated as unlikely. Other reasons might be measurement failures or installation errors in this deeper part of the soil profile.

Within this 22-year period, there were time periods with a good match between simulated and measured soil water contents and pressure heads as well as time spans with a distinct lower goodness of fit between modelling results and field observations (Figures 5–10). Due to these temporal variations in the simulation quality of the model at all field plots, we assumed that inadequate vGM-parameters were not the reason for these mismatches between simulation and measurements since from our point of view such inadequate vGM-parameters should have led to deviations between model calculations and observations throughout the total investigation period from 1993 to 2014.

In our study, an overestimation of RWU by the model in some soil compartments as reason for a mismatch between simulations and observations might be due to an overestimation of potential RWU, which depends on LAI and $ET_p$ as the upper threshold for soil water extraction. It is well-known that the application of different methods for assessing $ET_p$ can have an impact on the model performance of soil water flux models, e.g., [42]. However, an overestimation of potential RWU would lead to much more faster soil water withdrawal down to the wilting point especially from the upper soil compartments 0–30 and 30–60 cm as compared to corresponding measurements of soil water contents and pressure heads. However, this was observed in our study only in 1999 (Figures 5–10). Thus, we attributed these mismatches between simulated and measured model outputs to

- Errors in RWU-calculations in the summer half years due to incorrect description of root density distribution and calculation of crop specific rooting depth by the model.
- Measurement errors of the TRIME-probes despite a quality check and correction of the measured soil water contents using gravimetry.
- Differences between TRIME and tensiometers regarding measurement scale, precision and measuring principle.

The results of our study emphasized that long-term periods of soil hydrological experimental field data with high contrasting hydrometeorological conditions and variations in crop cover enable a more rigid evaluation of the model performance as compared to shorter periods with a lower variation of hydrometeorological conditions and crop cover.

## 5. Conclusions

With respect to the long-term stability of the soil hydrological measurement systems, time series of soil water contents measured by TRIME showed lower amounts of missing data in comparison with time series of pressure heads recorded by the tensiometers used in our study (Tables 4 and 5). Especially at 30 and 60 cm depth, there were longer periods without measured pressure head data due to drying out in summer and the necessity of a removal of the tensiometers during frost periods in winter (Figures 5–10, Table 5).

Simultaneous measurements of pressure heads by tensiometers and soil water contents by TRIME can be used for an a priori detection of errors in the time series in the case of contradictions between both measured state variables.

Despite a correction of the TRIME-measurements using soil water contents determined by gravimetry and a quality check, our results also suggested the existence of remaining temporary measurement errors of the TRIME-probes within this long-term period of observations. This substantiated the necessity of a thorough and real time quality check of the time series of automatic and continuous measured soil water contents and pressure heads as well. Such a quality check might be supported by the use of a soil water flux model for a further identification of erroneous and contradictory measurements of soil hydrological conditions.

Some of the mismatches between simulated and measured soil water contents and pressure heads in the summer half years were attributed to inadequate RWU-calculations. Thus, information about actual root density distribution and rooting depth in the field would allow a much more precise and consistent modelling of soil water withdrawal by RWU and an improvement of RWU-models. Furthermore, the use of measurement devices for the determination of actual evapotranspiration rates such as Eddy-Covariance might further support a thorough model validation. However, such devices are expensive, the operation is time and labor consuming and the necessity of management practices such as tillage, planting, fertilizer applications and harvest at agricultural field plots means an additional difficulty for the installation and operation of such Eddy-Covariance systems.

The combined use of TRIME and tensiometer can lead to different results with regard to the simulation quality of the soil water flux model. For example, model performance at Plot 3 for soil water contents at the soil compartment 30–60 cm in terms of IA was at 0.84 and $R^2$ at 0.63 (Table 5). Contrarily, model performance for corresponding pressure heads at 60 cm depth at Plot 3 was described by an IA 0.28 and $R^2$ at 0.11 (Table 5). Thus, a good fit between simulated and observed soil water contents does not necessarily result in a comparably good fit between corresponding calculated and measured pressure heads.

**Author Contributions:** M.W. carried out the modelling calculations, data quality check and wrote the paper. K.L. sampled and prepared the field data from different sources. D.S. was responsible for the operation and technical supervision of the meteorological and soil hydrological measurement systems. D.B. and W.M. were involved in the coordination and design of the field experiments and the crop rotations and contributed to writing and editing of the paper.

**Funding:** This study was financially supported by the German Federal Ministry of Consumer Protection, Food and Agriculture and the Ministry of Agriculture, Environmental Protection and Regional Planning of the Federal State of Brandenburg (Germany).

**Acknowledgments:** The authors want to thank our colleagues at ZALF Experimental Station and others for conducting the field experiments.

**Conflicts of Interest:** The authors declare no conflict of interest.

**Data Availability:** The 22-year dataset is available via doi:10.4228/ZALF.DK.49. Additional information can obtained via doi:10.4228/ZALF.1992.167.

# Appendix A

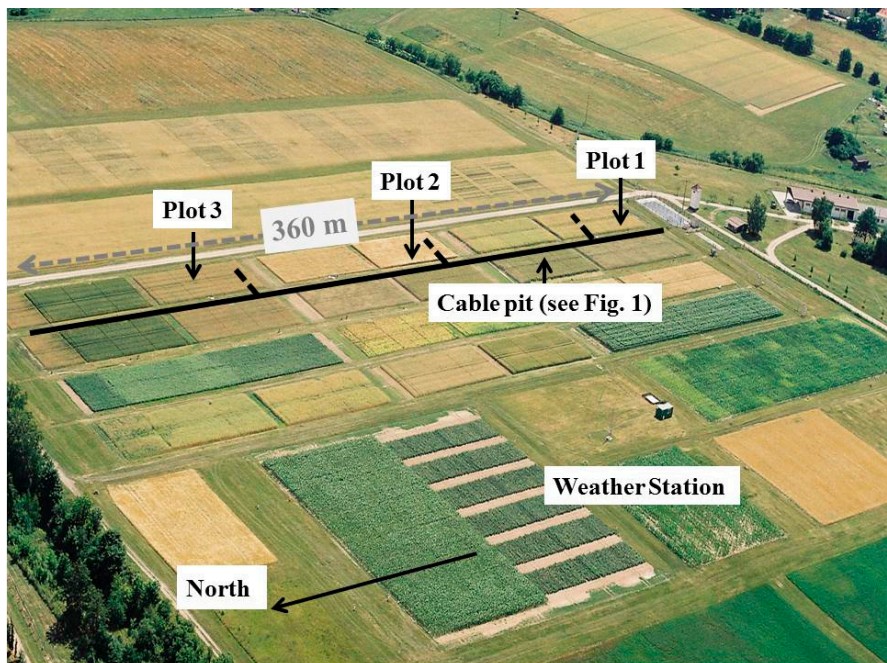

**Figure A1.** Location of the three field plots (Image by B. Zbell), from ([38], modified).

**Table A1.** Crop rotation, sowing date, harvest date and yield data (t ha$^{-1}$), Plot 1 ([38], modified).

| Year | Crop | Sowing (day.month.year) | Harvest (day.month.year) | Observed Yield | Simulated Yield |
|---|---|---|---|---|---|
| 1993 | Sugar beet | 26.04.93 | 06.10.93 | 8.455 | 9.955 |
| 1993/1994 | Winter wheat | 15.10.93 | 29.07.94 | 4.516 | 6.150 |
| 1994/1995 | Winter barley | 26.09.94 | 21.07.95 | 5.652 | 7.988 |
| 1995/1996 | Winter rye | 02.10.95 | 21.08.96 | 6.812 | 8.423 |
| 1996 | Oil redish (winter catch crop) | 05.09.96 | - | - | - |
| 1997 | Sugar beet | 03.04.97 | 23.09.97 | 11.639 | 10.789 |
| 1997/1998 | Winter wheat | 08.10.97 | 27.07.98 | 3.490 | 5.120 |
| 1999 | Lucerne-clover-grass-mix | - | - | - | - |
| 2000 | Lucerne-clover-grass-mix | - | - | - | - |
| 2000/2001 | Winter wheat | 20.09.00 | 27.07.01 | 6.678 | 6.695 |
| 2001/2002 | Winter barley | 21.09.01 | 06.06.02 | - | - |
| 2002/2003 | Winter oilseed rape | 19.08.02 | 11.07.03 | 2.780 | 1.988 |
| 2003/2004 | Winter triticale | 21.09.03 | 31.07.04 | 6.360 | 8.123 |
| 2004/2005 | Winter rye | 23.09.04 | 02.08.05 | 6.270 | 6.195 |
| 2006 | Potatoes | 24.04.06 | 15.08.06 | 4.116 | 3.578 |
| 2007 | Sorghum | 23.05.07 | 27.09.07 | 10.112 | - |
| 2008 | Sorghum | 29.05.08 | 10.09.08 | 5.492 | - |
| 2008/2009 | Winter triticale | 24.09.08 | 23.06.09 | 9.706 | 8.408 |
| 2009 | Lucerne-clover-grass-mix | 03.07.09 | 25.08.09 | 3.764 | 3.212 |
| 2009 | " | - | 20.10.09 | 1.155 | 2.112 |
| 2010 | " | - | 26.0510 | 6.421 | 3.762 |
| 2010 | " | - | 16.07.10 | 2.649 | 2.223 |
| 2010 | " | - | 05.10.10 | 4.741 | 3.512 |
| 2011 | Lucerne-clover-grass-mix | - | 21.04.11 | - | - |
| 2011 | Silage maize | 02.05.11 | 14.09.11 | 23.623 | 19.989 |
| 201/2012 | Winter rye for green biomass | 26.09.11 | 23.05.12 | 8.461 | 5.678 |
| 2012 | Sorghum | 24.05.12 | 18.09.12 | 10.339 | - |
| 2012/2013 | Winter triticale for green biomass | 28.09.12 | 24.06.13 | 10.939 | 8.173 |
| 2013 | Lucerne-clover-grass-mix | 08.07.13 | 10.09.13 | 2.564 | 3.121 |
| 2014 | " | - | 14.05.14 | 7.443 | 4.123 |
| 2014 | " | - | 02.07.14 | 5.869 | 3.176 |
| 2014 | " | - | 05.08.14 | 2.307 | 2.221 |
| 2014 | " | - | 09.10.14 | 3.561 | 2.456 |

**Table A2.** Crop rotation, sowing date, harvest date and yield (t ha$^{-1}$), Plot 2 ([38], modified).

| Year | Crop | Sowing (day.month.year) | Harvest (day.month.year) | Observed Yield | Simulated Yield |
|---|---|---|---|---|---|
| 1992/1993 | Sugar beet | 26.04.93 | 06.10.93 | 9.467 | 12.123 |
| 1993/1994 | Winter wheat | 15.10.93 | 29.07.94 | 3.245 | 5.123 |
| 1994/1995 | Winter barley | 26.09.94 | 21.07.95 | 3.020 | 4.134 |
| 1995/1996 | Winter rye | 02.10.95 | 21.08.96 | 2.450 | 4.123 |
| 1996 | Yellow mustard (winter catch crop) | 05.09.96 | - | - | - |
| 1997 | Sugar beet | 03.04.97 | 23.09.97 | 11.762 | 11.706 |
| 1997/1998 | Winter wheat | 08.10.97 | 27.07.98 | 1.430 | 4.123 |
| 1999 | Lucerne-clover-grass-mix | - | - | - | - |
| 2000 | " | - | - | - | - |
| 2000/2001 | Winter rye | 17.09.00 | 25.07.01 | 6.580 | 7.826 |
| 2002 | Peas | 11.04.02 | 16.07.02 | 2.710 | - |
| 2002/2003 | Winter barley | 16.09.02 | 14.07.03 | 3.600 | 6.720 |
| 2003/2004 | Winter oilseed rape | 20.08.03 | 23.07.04 | 4.280 | 2.315 |
| 2004/2005 | Winter wheat | 23.09.04 | 01.08.05 | 4.610 | 6.723 |
| 2006 | Silage maize | 02.05.06 | 12.09.06 | 5.820 | 6.928 |
| 2007 | Winter rye for green biomass | 19.09.06 | 24.04.07 | - | - |
| 2007 | Sorghum | 23.05.07 | 27.09.07 | 7.596 | - |
| 2007/2008 | Winter rye for green biomass | 02.10.07 | 28.05.08 | 5.710 | 3.935 |
| 2008 | Sorghum | 29.05.08 | 10.09.08 | 5.264 | - |
| 2008/2009 | Winter triticale | 24.09.08 | 23.06.09 | 9.365 | 8.322 |
| 2009 | Lucerne-clover-grass-mix | 03.07.09 | 25.08.09 | 4.391 | 3.213 |
| 2009 | " | - | 20.10.09 | 1.248 | 2.111 |
| 2010 | " | - | 26.05.10 | 6.374 | 3.517 |
| 2010 | " | - | 15.07.10 | 2.122 | 2.213 |
| 2010 | " | - | 05.10.10 | 5.166 | 3.613 |
| 2011 | Lucerne-clover-grass-mix | - | 21.04.11 | - | - |
| 2011 | Silage maize | 02.05.11 | 14.09.11 | 20.275 | 17.845 |
| 2011/2012 | Winter rye for green biomass | 26.09.11 | 23.05.12 | 6.760 | 3.213 |
| 2012 | Sorghum | 24.05.12 | 18.09.12 | 8.639 | |
| 2012/2013 | Winter triticale | 28.09.12 | 24.06.13 | 7.248 | 7234 |
| 2013 | Lucerne-clover-grass-mix | 08.07.13 | 10.09.13 | - | - |
| 2014 | " | - | 14.05.14 | 5.611 | 3.231 |
| 2014 | " | - | 02.07.14 | 6.052 | 3.455 |
| 2014 | " | - | 05.08.14 | 2.712 | 2.221 |
| 2014 | " | - | 09.10.14 | 3.259 | 2.345 |

**Table A3.** Crop rotation, sowing date, harvest date, and yield (t ha$^{-1}$), Plot 3 ([38], modified).

| Year | Crop | Sowing (day.month.year) | Harvest (day.month.year) | Observed Yield | Simulated Yield |
|---|---|---|---|---|---|
| 1993 | Sugar beet | 26.04.93 | 06.10.93 | 15.427 | 13.896 |
| 1993/1994 | Winter wheat | 15.10.93 | 29.07.94 | 4.797 | 6.344 |
| 1994/1995 | Winter barley | 26.09.94 | 21.07.95 | 5.681 | 7.123 |
| 1995/1996 | Winter rye | 02.10.95 | 21.08.96 | 5.135 | 5.344 |
| 1996 | Phacelia | 05.09.96 | - | - | - |
| 1997 | Sugar beet | 03.04.97 | 23.09.97 | 12.419 | 11.762 |
| 1997/1998 | Winter wheat | 08.10.97 | 27.07.98 | 4.368 | 7.289 |
| 1999 | Lucerne-clover-grass-mix | - | - | - | - |
| 2000 | " | - | - | - | - |
| 2000/2001 | Winter rye | 17.09.00 | 25.07.01 | 7.965 | 8.242 |
| 2002 | Potatoes | 22.04.02 | 15.08.02 | 10.382 | 7.675 |
| 2002/2003 | Winter wheat | 20.09.02 | 21.07.03 | 4.700 | 6.123 |
| 2003/2004 | Winter barley | 14.09.03 | 08.07.04 | 6.650 | 8.127 |
| 2004/2005 | Winter oil seed rape | 20.08.04 | 18.07.05 | 3.410 | 1.945 |
| 2005/2006 | Winter triticale | 23.09.05 | 18.07.06 | 4.010 | 6.981 |
| 2006/2007 | Winter rye for green biomass | 19.09.06 | 21.05.07 | - | - |
| 2007 | Sorghum | 23.05.07 | 10.09.07 | 11.231 | - |
| 2007/2008 | Winter rye for green biomass | 02.10.07 | 30.03.08 | - | - |
| 2008 | Silage maize | 24.04.08 | 05.09.08 | 10.790 | 7.981 |
| 2008/2009 | Winter rye for green biomass | 24.09.08 | 30.03.09 | - | - |
| 2009 | Silage maize | 30.04.09 | 11.09.09 | 21.979 | 17.896 |
| 2009/2010 | Winter rye for green biomass | 24.09.09 | 30.03.10 | - | - |
| 2010 | Silage maize | 30.04.10 | 18.09.10 | 15.330 | 10.234 |
| 2010/2011 | Winter rye for green biomass | 24.09.10 | 21.04.11 | - | - |
| 2011 | Silage maize | 02.05.11 | 14.09.11 | 22.684 | 17.331 |
| 2011/2012 | Winter rye for green biomass | 26.09.11 | 11.04.12 | - | - |

**Table A3.** *Cont.*

| Year | Crop | Sowing (day.month.year) | Harvest (day.month.year) | Observed Yield | Simulated Yield |
|---|---|---|---|---|---|
| 2012 | Silage maize | 25.04.12 | 17.09.12 | 21.504 | 16.123 |
| 2012/2013 | Winter rye for green biomass | 28.09.12 | 30.03.13 | - | |
| 2013 | Silage maize | 26.04.13 | 12.09.13 | 16.412 | 12.325 |
| 2014 | Winter rye for green biomass | 25.09.13 | 14.04.14 | - | - |
| 2014 | Silage maize | 30.04.14 | 18.09.14 | - | - |

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
