# Peer review of "Simulation of Long-Term Soil Hydrological Conditions at Three Agricultural Experimental Field Plots Compared with Measurements"

_water, doi:10.3390/w11050989_

Reviewer 1 Report

This study is within the scope of the journal. The paper is based on intensive field observations on soil water content. The authors compared the differences between the estimated and the observed soil water content with two methods by tensiometer and TRIME-probes. In general, I'm supportive of publication of this paper. However, prior to publication, some concerns need to be addressed.

1.    Please added the reference of the method used in calculating ETP, which is important variable in the estimation of this study. Also, the authors need to specify all the parameters value such as ext_coeff in the paper. In addition, how about the sensitivity of ETP on the estimated soil water content need to be illustrate since there many different kinds of methods can be used to estimate ETP.

2.   The pressure head method seems have less good performance for plot 1 at the deeper depth compared with plot 2 and 3, the reasons might be the differences in soil physical property, the authors can give more explanations or discussions.

Other small modifications are also needed to improve the paper, such as:

1.     Add a legend for each figure

2.  Refine the conclusion section to better match the objective proposed in the introduction section

Author Response

Response to Reviewer 1 Comments

We have to thank for the constructive and helpful comments of Reviewer 1 for an improvement of our submitted paper. For this purpose, we have now added coloured figures to enhance the quality of our figures and we have tried to follow all recommendations and comments of Reviewer 1. Our changes in the revised version of our manuscript are marked in red color.

 Comment 1: Please added the reference of the method used in calculating ETp, which is important variable in the estimation of this study. Also, the authors need to specify all the parameters value such as ext_coeff in the paper. In addition, how about the sensitivity of ETP on the estimated soil water content need to be illustrate since there many different kinds of methods can be used to estimate ETp.

 Response 1: We agree, that there exist a lot of approaches for the calculation of ETp.  However, in most of the process-based agroecosystem models, ETp is estimated using methods based on the Penman- or Penman-Monteith-equation. The  references for calculating ETp were added in the revised version of the paper (see line 91-92). Furthermore, we have rearranged chapter 2.1 (Line 99-102) and the values of ext_coeff (equation (2) were summarized in line 225-226. In addition, we include now ETp-rates in Figure 4 (see also line 266-269).  A discussion about the impact of ETp on model performance was added in line 397-399 and in line 578-587 in the revised version of our paper.

 Comment 2: The pressure head method seems have less good performance for plot 1 at the deeper depth compared with plot 2 and 3, the reasons might be the differences in soil physical property, the authors can give more explanations or discussions.

 Response 2: A discussion about this potential impact of different soil physical properties on model performance for soil water pressure heads in the lower part of the soil profile can be found in line 556-569 of the revised version of the paper.

Comment 3: Other small modifications are also needed to improve the paper, such as:

1.  Add a legend for each figure

2.  Refine the conclusion section to better match the objective proposed in the introduction section

Response 3: Legends were added for each figure. However, from our point of view, conclusions should be drawn depending on the findings of the study

Reviewer 2 Report

I think a better title would increase the attractiveness of the paper. Something like: Simulation of long-term soil hydrological conditions compared to measurements. If more attention was paid to the impact of crop management, the paper would be more interesting. Please, explain why these three plots were selected for investigation and comparison. What is the difference between them? Is the variation large enough to make them relevant for the investigation?

For other comments see the manuscript.

Author Response

Response to Reviewer 2 Comments

We have to thank for the constructive and helpful comments of Reviewer 2 for an improvement of our submitted paper. For that purpoese, we have now added coloured figures to enhance the quality of our papers. Our changes in the revised version of our manuscript are marked in red color. More details are in the responses to the comments of Reviewer 2.

 Comment 1: I think a better title would increase the attractiveness of the paper. Something like: Simulation of long-term soil hydrological conditions compared to measurements.

 Response 1: We have changed the title in the revised version of our paper

 Comment 2: If more attention was paid to the impact of crop management, the paper would be more interesting.

 Response 2: It is well-known that soil water availability for root water uptake is one of the most limiting factors for crop growth under rainfed conditions. This could have been observed in Europe in dry years such as 2003 or 2018. However, an impact of the different crop rotations established at the three field plots on soil water balance is illustrated in Figure 3-4 and a discussion about this impact can be found in line 531-544 of the revised paper. From our point of view, an additional more detailed discussion of the impact of crop management would enlarge the paper too much since the focus of our study was soil water balance and we simulated only water limited crop production. However, this impact of crop management should be discussed and analyzed in more detail in a following paper taking into account the results of our submitted paper.

 Comment 3: Please, explain why these three plots were selected for investigation and comparison. What is the difference between them? Is the variation large enough to make them relevant for the investigation?

 Response 3: We have added additional information in line 154-162 in the revised version of the manuscript. From our point of view, these variations regarding the applied cropping systems and cultivated crop types make the plots relevant for the investigation (see also Table A1-A3).   In addition, we followed the comments and recommendations of Reviewer 2 directly added in the first version of the submitted paper.

Reviewer 3 Report

Review of "Measured and Simulated Long-Term Soil Hydrological Conditions at three Agricultural 3 Experimental Field Plots" by Wegehenkel et al.

This paper addresses an important issues with wide ranging implications--challenges in numerical modeling of the observed state of the vadose zone. All aspects of the methods appear appropriate. However, I am concerned that no literature gap is identified in the introduction or abstract, no hypothesis is identified to be tested, and the authors do not argue that their results are in any way novel.

Author Response

Response to Reviewer 3 Comments

Again, we have to thank for the constructive and helpful comments of Reviewer 3 for an improvement of our submitted paper. For this improvement, we have now added coloured figures. Our changes in the revised version of our manuscript are marked in red color. From our point of view, model validation using experimental field data is still a necessary contribution for an analysis of limitations and further improvement of simulation models, despite the fact that the comparison of measured and simulated model outputs is an already established scientific practice and,  therefore, not really new. However, to answer the main question by Reviewer 3 and the Associate Editor regarding the novelty of our study, we added some new references with measurement set ups (TDR and daily time step) similar to those applied in our study and with the same focus on model validation. The time series of measured soil hydrological conditions used in these references [see 13, 16-24] were significantly shorter and with a lower variation in crop cover as compared to our submitted and revised paper (see line 63-67 and line 78-84).  More details are in the responses to the comments of Reviewer 3.

 Comment 1: This paper addresses an important issues with wide ranging implications--challenges in numerical modeling of the observed state of the vadose zone. All aspects of the methods appear appropriate. However, I am concerned that no literature gap is identified in the introduction or abstract, no hypothesis is identified to be tested, and the authors do not argue that their results are in any way novel.

 Response 1: We have tried to clarify this in line 63-66 and line 79-82 in the revised version of our manuscript. From our point of view, model validation studies based on such continuous long-term soil hydrological data sets with daily timesteps using different measurement techniques (Gravimetry, TDR and Tensiometer) are rare, as far as the authors know. From our point of view, such coherent data sets offer new opportunities for, e.g., model validation and model improvement as compared with other recent studies. We have added some additional new references in the revised version of our paper for a further substantiation of this novel aspect of our study (Line 63-67 and line 78-82). Especially the application of process-based agroecosystem models for the estimation of the impact of climate change on, e.g., evapotranspiration, soil water storage and  groundwater recharge, requires model validation using long-term data to check the quality and consistence of the model predictions for soil water balance over such  long-time periods. The results of our study suggested that shorter validation periods with measured soil water contents and observed pressure heads might lead to a misinterpretation of model performance in some cases (see line 571-578 and Figure 5-10).  For example, simulated soil water contents at the 30-60 cm compartment, Plot 2, showed a good fit to the measured ones from 2008 to 2014, but from 1993 to 2006, higher deviations between calculated and observed soil water contents could have been observed (Figure 7c-8c).

Round  2

Reviewer 2 Report

Thank you for clarifying some points in the new version of the paper.  I am fully satisfied with the responses to my questions.